

# Alpine topography of the Gamburtsev Subglacial Mountains, Antarctica, mapped from ice sheet surface morphology

Edmund J. Lea[1], Stewart S.R. Jamieson[1], Michael J. Bentley[1]

[1]Department of Geography, Durham University, South Rd, Durham, DH1 3LE, UK

*Correspondence to*: Edmund J. Lea (edmund.j.lea@durham.ac.uk)

**Abstract.** Landscapes buried beneath the Antarctic Ice Sheet preserve information about the geologic and geomorphic evolution of the continent both before and during the wide-scale glaciation that began roughly 34 million years ago. Throughout this time, some areas of the ice sheet have remained cold-based and non-erosive, preserving ancient landscapes remarkably intact. The Gamburtsev Subglacial Mountains in central East Antarctica are one such landscape, maintaining evidence of

tectonic, fluvial and glacial controls on their distinctly alpine morphology. The central Gamburtsevs have previously been surveyed using airborne ice-penetrating radar, however, many questions remain as to their evolution and their influence on the East Antarctic Ice Sheet, including where in the region to drill for a 1.5-million-year-long 'Oldest Ice' core. Here, we derive new maps of the planform geometry of the Gamburtsev Subglacial Mountains from satellite remote sensing datasets of the ice sheet surface, based on the relationship between bed roughness and ice surface morphology. Automated and manual

approaches to mapping were tested and validated against existing radar data and elevation models. Manual mapping was more effective than automated approaches at reproducing bed features observed in radar data, but a hybrid approach is suggested for future work. The maps produced here show detail of mountain ridges and valleys on wavelengths significantly smaller than the spacing of existing radar flightlines, and mapping has extended well beyond the confines of existing radar surveys. Morphometric analysis of the mapped landscape reveals that it constitutes a preserved (> 34 Ma) dendritic valley network,

with some evidence for modification by topographically confined glaciation prior to ice sheet inception. The planform geometry of the landscape is a significant control on locations of basal melting, subglacial hydrological flows, and the stability of the ice sheet over time, so the maps presented here may help to guide decisions about where to search for Oldest Ice.

## 1 Background and rationale

Ice that remains frozen to its bed largely protects subglacial topography from erosion, preserving landscapes formed millions

of years ago under climatic conditions and/or ice sheet configurations drastically different to those of the present day (e.g. Jamieson et al., 2005; Young et al., 2011; Rose et al., 2013; Paxman et al, 2018; Franke et al., 2021). Subglacial landscapes can therefore provide unique records of glacial and preglacial geomorphic processes. Their importance as controls on ice sheet initiation, evolution, and potential collapse has also been recognised (DeConto and Pollard, 2003; Sugden and John, 1976; Mercer, 1978), and a key focus in recent years has been the production of gridded subglacial topographic datasets that can be





used as boundary conditions for ice sheet models (Fretwell et al., 2013; Morlighem et al., 2020; Frémand et al., 2022). Standard methods using airborne ice penetrating radar (or radio echo sounding; RES) to measure ice thickness, and hence derive bed elevations, are effective at producing vertical profiles, but rely on interpolation techniques (e.g. Morlighem et al., 2011) or the use of coarser-resolution or satellite-derived gravity data (Fretwell et al., 2013) to fill gaps between transects (flightlines). These gaps are rarely smaller than 5-10 km, and often much larger (Fretwell et al., 2013; Pritchard et al., 2014), hence the

planform (horizontal) structures of subglacial landscapes may be poorly known at significant scales, even in areas where RES coverage is good.

An approach to mapping subglacial landscapes with the potential to address this issue is the use of ice sheet surface morphology (Le Brocq et al., 2008; Ross et al., 2014; Chang et al., 2016), which records the fingerprint of subglacial topography due to its influence on ice flow (Rémy and Minster, 1997). Changes in slope, or curvature, of the ice surface can be extracted from radar-

based satellite imagery (e.g. MODIS and RADARSAT image mosaics; Scambos et al., 2007; Jezek et al., 2013) and digital elevation models (DEMs; e.g. Bamber et al., 2009; Howat et al., 2019), and used to map out the planform geometry of subglacial ridges and valleys over which the modern ice sheet flows. This information can be used to fill gaps between RES surveys (e.g. Ross et al., 2014), or give a preliminary indication of the landscape structure in areas with few existing data (e.g. Le Brocq et al., 2008; Jamieson et al., 2016). Moreover, recent high-resolution datasets such as the Reference Elevation Model

of Antarctica DEM (Howat et al., 2019) and the RADARSAT-1 Antarctic Mapping Project image mosaic (Jezek et al., 2013), offer the possibility to enhance knowledge of the fine-scale topographic structure of ice sheet beds even in areas previously well-surveyed.

One such region is the enigmatic landscape of the Gamburtsev Subglacial Mountains (GSM), in central East Antarctica. Bounded by the Lambert Graben to the North, the South Pole Basin to the South, and a complex system of linear faults

(Ferraccioli et al., 2011) to East and West (Fig. 1), the GSM are a high-relief, alpine mountain range (Bo et al., 2009; Rose et al., 2013) entirely submerged beneath Dome A, the highest point of the Antarctic Ice Sheet. Prior to the internationally collaborative Antarctica's Gamburtsev Province (AGAP) project during the International Polar Year of 2007–2009 (Bell et al., 2011; Ferraccioli et al., 2011), very little was known about their structure. Analysis of the RES data collected by the AGAP surveys revealed a dendritic network of subglacial valleys surrounded by large mountain massifs, bearing evidence of multiple

stages of glacial modification (Rose et al., 2013). The relatively high level of detail achieved by the AGAP survey does not persist outside of the survey area, but the few existing measurements suggest that the surrounding foothills of the GSM may be much more extensive. The surveys provide a clear benchmark against which to validate interpretations made from ice surface mapping (c.f. Ross et al., 2014; Jamieson et al., 2016), which may also expand detail of the landscape beyond the surveyed region, and in between AGAP flightlines. The high bed relief (Bell et al., 2011; Rose et al., 2013), relatively thin ice,

and slow ice flow (Mouginot et al., 2019) over the GSM produce surface expressions that can be mapped to infer landscape structure.





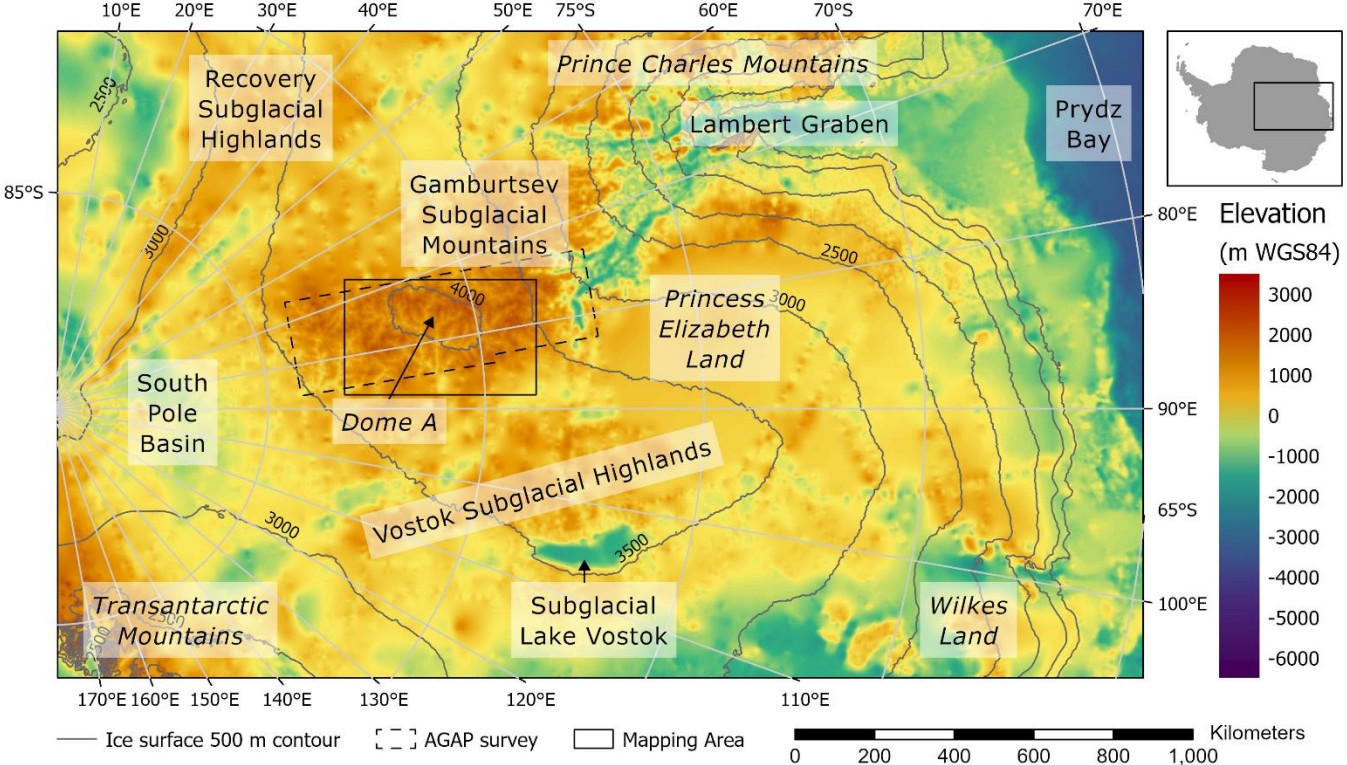

**Figure 1 – The regional subglacial topography of central East Antarctica from Bedmap2 (Fretwell et al., 2013), including placenames mentioned in the text. Names in italics = surficial/subaerial features, names in regular typeface = bed features. The area of interest for this study (Fig. 2, Fig. 5, Fig. 9) is indicated by the solid box. AGAP = Antarctica's Gamburtsev Province.**

The aim of this study is to understand the long-term landscape and ice sheet evolution in the Gamburtsev Subglacial Mountains using information about the planform geometry of the landscape mapped from ice surface datasets. Our objectives are: (1) to test and compare methods for delineating subglacial ridges and valleys from ice surface datasets: achieved by processing remotely-sensed data to derive ice surface curvature, and using manual and automated techniques to map changes in slope; (2) to map the subglacial valley and ridge networks of the GSM, using the methods tested in step 1 with datasets of the Reference Elevation Model of Antarctica (REMA) ice surface DEM (Howat et al., 2019) and the RADARSAT-1 Antarctic Mapping Project (RAMP) image mosaic version 2 (Jezek et al., 2013), both at 200 m spatial resolution; (3) to evaluate the mapped networks against existing bed elevation models and RES data taken from AGAP surveys (Bell et al., 2011; Ferraccioli et al., 2011) and BedMachine Antarctica (Morlighem et al., 2020); (4) to analyse the morphometry of the mapped networks, using metrics including valley length, orientation, and spacing; and (5) to interpret the mapped landscape with respect to geomorphic processes and ice sheet behaviour, by assessing the role of different processes (glacial, fluvial, tectonic) in its evolution, and its implications for the past and present evolution of the Antarctic Ice Sheet.



## 1.1 Study area: The Gamburtsev Subglacial Mountains (GSM)

The Gamburtsev Subglacial Mountains (Fig. 2) are, geologically, poorly understood, with their location and total submersion
beneath the high plateau of the East Antarctic Ice Sheet (Dome A) rendering them currently inaccessible for direct sampling
of bedrock. Their origin has been a persistent question since their discovery (Sorokhtin et al., 1959), with debate around the
cause and timing of uplift remaining largely unresolved (van de Flierdt et al., 2008; Block et al., 2009; Ferraccioli et al., 2011;
Heeszel et al., 2013; Paxman et al., 2016). Several hypotheses have been proposed, including geologically recent thermal uplift
associated with mantle hot spot activity (Sleep, 2006), ancient orogeny in a continental collision zone (Fitzsimmons, 2000,
2003; Block et al., 2009), crustal shortening in response to long-distance stress transmission (Veevers, 1994), and uplift
associated with rifting during continental breakup (Ferraccioli et al., 2011). Attempts have been made to date their formation
using detrital materials of likely ancestral GSM provenance from coastal and offshore sediment deposits in the Prince Charles
Mountains and Prydz Bay (Veevers and Saeed, 2008; Veevers et al., 2008; van de Flierdt et al., 2008; Gupta et al., 2022),
producing ages generally in the ranges ca. 1200–800 Ma and ca. 620–460 Ma (Veevers and Saeed, 2008; van de Flierdt et al.,
2008), with another potential period of activity ca. 700 Ma (Gupta et al., 2022). The landscape of the GSM has been
characterised as predominantly fluvial in origin (Rose et al., 2013), bearing evidence of a dendritic valley network organised
into discrete drainage basins, concave valley long-profiles, and some V-shaped valley forms (Bo et al., 2009; Rose et al., 2013).
Modelling based on low rates of erosion (c.f. Cox et al., 2010) suggests that the fluvial network of the GSM is no more than
230 Myr old (Paxman et al., 2016).

Evidence is also found for modification of the fluvial landscape by topographically confined, alpine-style glaciation, prior to
the formation of regional or continental-scale ice caps (Bo et al., 2009; Rose et al., 2013). Glacial landforms present in the
landscape, such as U-shaped troughs, overdeepenings, cirques, and hanging valleys, are indicative of this style of glaciation,
and incompatible with formation under modern ice sheet flow. A significant step-change in the benthic oxygen isotope ratio
is used to place continental ice sheet inception in Antarctica at the Eocene-Oligocene transition ca. 34 Ma, due to the link
between terrestrial ice volume and storage of heavy ($^{18}$O) isotopes (Coxall et al., 2005). It may therefore be inferred that phases
of cirque and alpine-style glaciation in the GSM were likely associated with periods of climatic cooling that predate this change
(Rose et al., 2013), though the exact timing of such episodes remains uncertain. There is growing evidence for fluctuating
glaciations in Antarctica during the late Eocene ca. 38–34 Ma (Van Breedam et al., 2022, and references therein), but some
authors argue for the existence of terrestrial ice as far back as the late Cretaceous, ca. 130 Ma, to explain changes in sea level
and ocean chemistry which occurred at that time (Stoll and Schrag, 1996; Miller et al., 2008).

The basal thermal regime is key to patterns of landscape erosion and preservation beneath the Antarctic Ice Sheet (Jamieson
et al., 2014), as significant glacial erosion is dependent on the occurrence of basal melting (Sugden and John, 1976). Models
suggest that ice in the GSM has remained cold-based since the early Oligocene (DeConto and Pollard, 2003; Jamieson et al.,
2010), leading to minimal rates of erosion. This conclusion is supported by low rates of offshore sedimentation in the
catchments down-ice from the GSM (Cox et al., 2010), and the preservation of the alpine landscape (Bo et al., 2009; Rose et

al., 2013). Bright reflections in AGAP RES profiles indicate that meltwater is present in the bottom of some overdeepened valleys of the GSM where ice flow is slow, but confirm that on peaks and valley sides, basal ice remains frozen to the bed (Bell et al., 2011; Wolovick et al., 2013; Creyts et al., 2014).

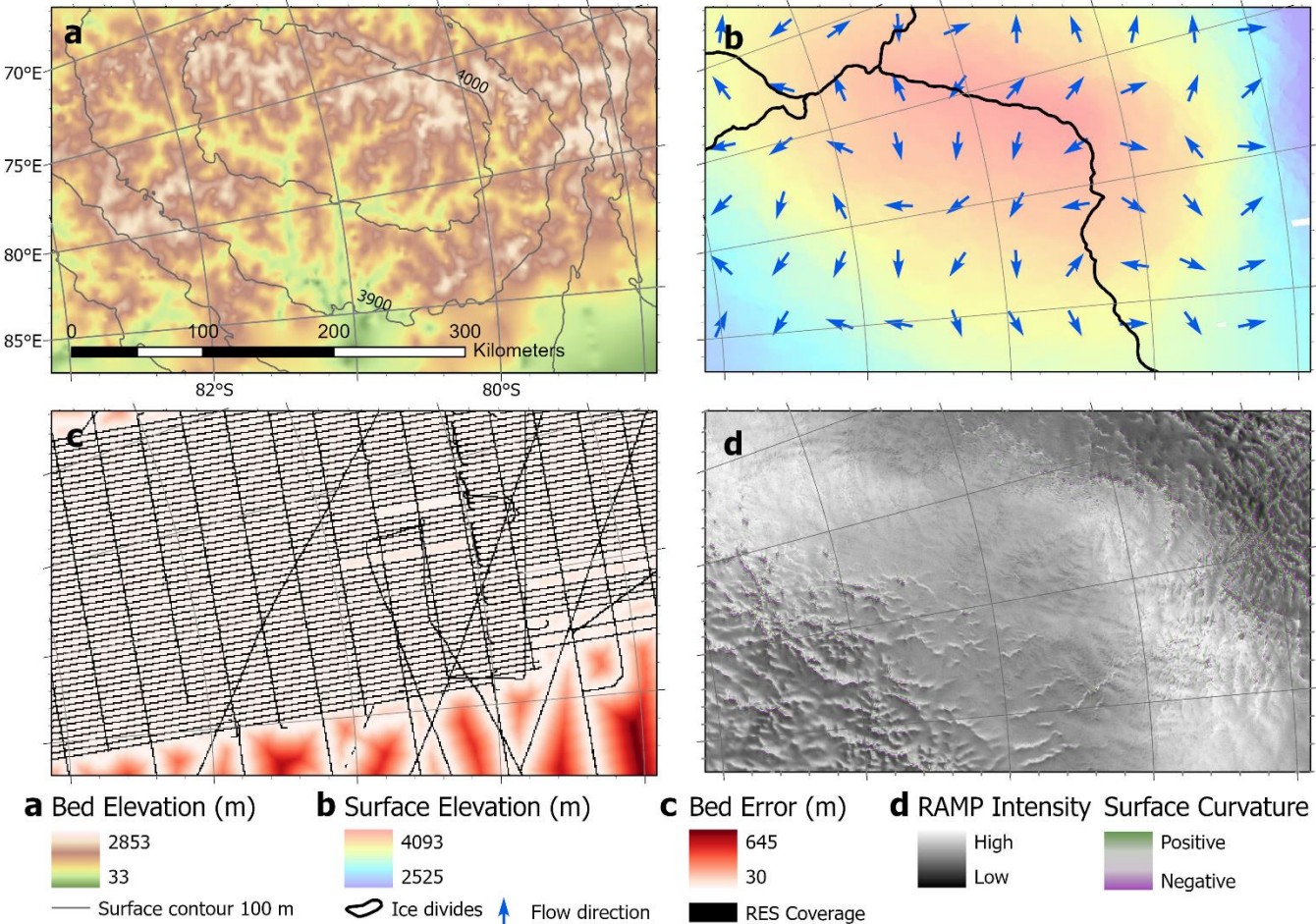

Figure 2 – **Area of the Gamburtsev Subglacial Mountains being targeted for mapping in this study. Bed elevation and errors from BedMachine Antarctica (Morlighem et al., 2020). Surface elevation from Reference Elevation Model of Antarctica (Howat et al., 2019). Flow vectors from MEaSUREs flow velocity (Mouginot et al., 2019). Ice divides from Zwally et al. (2012). Radio echo sounding (RES) flightlines from Fretwell et al. (2013). Satellite image from RADARSAT-1 Antarctic Mapping Project (RAMP) image mosaic version 2 (Jezek et al., 2013).**

## 2 Methods

We used satellite-derived datasets relating to ice surface morphology, in isolation, to map the planform geometry of the central part of the Gamburtsev Subglacial Mountains. We tested both automated and manual approaches to digitising changes in ice surface slope, inferred to represent subglacial valleys and ridges, and assessed the results against existing radar observations (Bell et al. 2011; Ferraccioli et al., 2011), and bed elevation model (Morlighem et al., 2020). We subsequently analysed the





morphometric characteristics of the revealed valley and ridge networks, in order to investigate the nature of the subglacial landscape. The two principal data products used were: (1) the Reference Elevation Model of Antarctica (REMA), a high-resolution, continental-scale digital elevation model (DEM) constructed using stereophotogrammetry from commercial optical satellite imagery (Howat et al., 2019); and (2) the RADARSAT-1 Antarctic Mapping Project (RAMP) AMM-1 Synthetic Aperture Radar (SAR) image Mosaic of Antarctica, Version 2, representing the radar backscatter intensities recorded by the SAR sensor (Jezek et al., 2013). Both products were used in their 200 m spatial resolution versions for consistency.

**2.1 Reference Elevation Model of Antarctica (REMA)**

In order to more easily identify subtle changes in surface morphology during mapping, curvature, or the rate of change of slope, was calculated from the REMA DEM using the "Surface Parameters" tool in ArcGIS Pro 2.8.2 (c.f. Rémy and Minster, 1997; Le Brocq et al., 2008; Ross et al., 2014). Three distinct versions of surface curvature can be calculated using this tool: (a) profile curvature, which records the rate of change of slope in the direction of the greatest slope at each location (the along-slope direction); (b) plan curvature, which is the same quantity in the perpendicular direction (across-slope); (c) standard (or mean) curvature, which is a mixture of the two. In each case, curvature is calculated from a neighbourhood surrounding each pixel, with the size of the neighbourhood defined by a distance which can be varied to suit the wavelength of variability in the surface. Increasing the neighbourhood distance can reduce the impact of short-wavelength noise, though it may also result in smoothing of sharp contrasts in the data. Several different neighbourhood distances were trialled, resulting in curvature rasters with varying degrees of definition and noise (Fig. 3). Based on a visual comparison of mean curvature outputs, it was judged that a neighbourhood distance of 1000 m provided a curvature dataset with an appropriate balance between minimising noise and avoiding blurring of surface features.



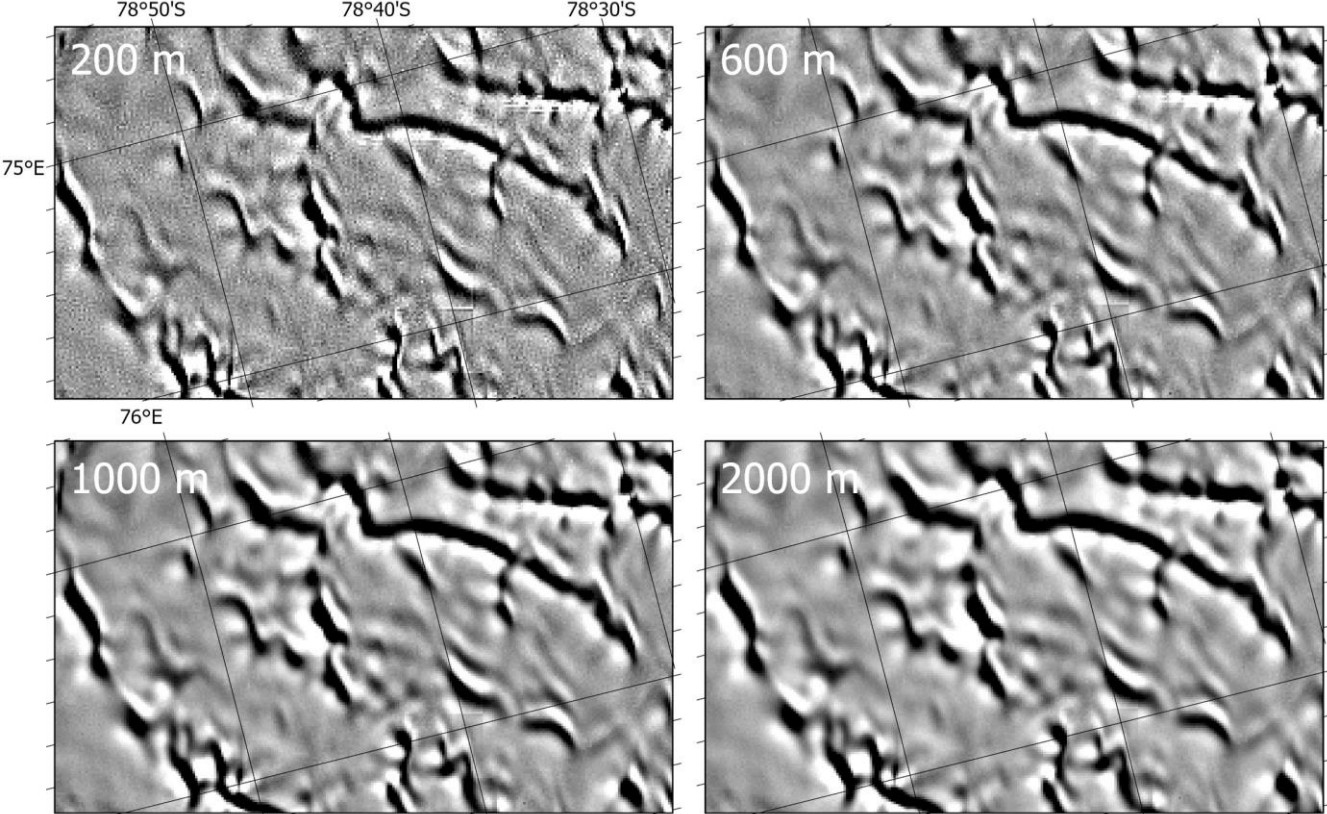

**Figure 3 – Extracts showing mean curvature of REMA surface elevation calculated using different neighbourhood distances (indicated top-left of each frame). The gradient varies, due to the differences in range produced using different distances. The 1000 m version (bottom left) was used in all further mapping and analyses.**

All three curvature products are useful for mapping, as they make it easier to identify the position of greatest slope in a step-change surface feature, represented by the transition from a curvature minimum to a curvature maximum (Fig. 4). For the curvature products used here, a positive value represents convexity (i.e. surface curved upwards), and a negative value represents concavity (i.e. surface curved downwards).



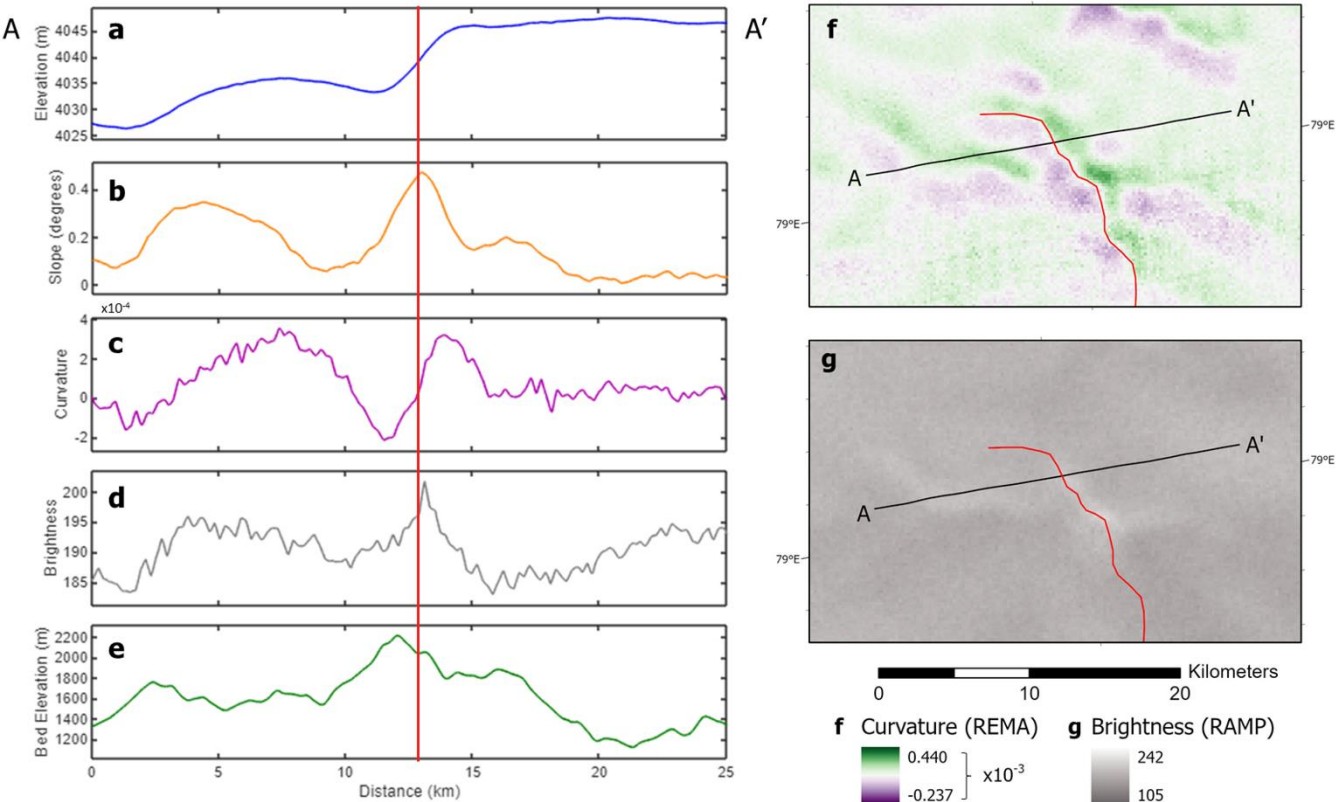

**Figure 4 – Profile (A-A') across a typical ice surface feature (red line), demonstrating the correspondence between the location of a bed ridge and its surface expression in the two datasets used for mapping: a) Reference Elevation Model of Antarctica (REMA; Howat et al., 2019) surface elevation; b) REMA surface slope; c) and f) REMA surface mean curvature (second derivative); d) and g) RADARSAT-1 Antarctic Mapping Project version 2 (RAMP; Jezek et al., 2013) brightness (adjusted backscatter intensity); e) Radio Echo Sounding bed elevation, Antarctica's Gamburtsev Province (AGAP) survey (Bell et al., 2011; Ferraccioli et al., 2011; Corr et al., 2020).**

## 2.2 RADARSAT-1 Antarctic Mapping Project (RAMP)

The pixel values for the RAMP image mosaic represent a qualitative measure of the intensity of radar backscatter (Jezek et al., 2013). The backscatter intensity is useful because it depends in part on the slope angle of the ice surface – a flatter surface leads to more signal being reflected directly back to the sensor, and hence a brighter backscatter value (Fig. 4).

## 2.3 Automated mapping

The RAMP image mosaic and the REMA mean curvature product were used as inputs to a multi-step automated mapping procedure. First, an adaptive binary threshold was applied to simplify the input data into a binary mask of "high" and "low" intensity/curvature regions, while accounting for differences in contrast across the study area. Then, an edge-detection algorithm with a directional input was used, to identify and categorise transitions in the binary image as linear features representing subglacial ridges and valleys. Additional pre- and post-processing procedures were used to smooth the data and





reduce the impact of noise, with slightly different processing steps required for each dataset[1]. InSAR phase-based ice velocities
(Mouginot et al., 2019), resampled at 200 m spatial resolution, provided the directional input for the edge-detection step.

## 2.4 Manual mapping

Manual mapping was conducted using a GIS, with changes in slope being manually traced as vector line features in a
geodatabase, with reference to the RAMP image mosaic and all three versions of REMA curvature. An important difference
between this approach and the automated procedure was that the manual mapping was conducted as a deliberately interpretive
process. Ridges and valleys were digitised separately, distinguished from one another both by the local flow direction with
regard to changes in curvature/intensity, and by the spatial relationships between features. Based on the existing knowledge
from RES surveys (Bo et al., 2009; Rose et al., 2013), and initial impressions formed when examining the datasets used for
mapping, it was assumed that the planform geometry being mapped would broadly resemble that of an alpine mountain
landscape originally shaped by fluvial erosion. As a result of this assumption, inferences could be made which were not feasible
to automate, particularly connections between features, such that isolated ridge or valley lines were joined – where small gaps
existed – to depict the likely structures of mountain chains and drainage networks. The results from automated mapping suggest
that this assumption of a network of valleys and ridges is a reasonable one.

Because they were represented more prominently, ridges in a given area were digitised first, followed by valleys. Following a
similar principle, the most obvious features of each type were traced first, then subtler features were picked out, especially
where these connected with established ridges or valleys. All digitised features were subsequently treated equally, with some
exceptions: in places, a connecting feature was inferred with no indication of its presence in any of the mapping data, or (in
the case of valleys) two possible connections were marked where there was ambiguity. Such cases mostly arose because the
assumption that the landscape would constitute a logical fluvial palaeo-drainage network, if true, occasionally required
drainage routes or drainage divides that were not observed. These features were marked as "low confidence" and were later
excluded from the dataset when performing some of the morphometric analyses. In the case of valleys, each was digitised from
its upper end towards its terminus (usually where it joined another valley), such that the direction of each line would reflect
the most likely direction of ice-free palaeo-drainage. Determination of valley direction constituted the only instance in which
existing bed models were referred to during mapping, as the theoretical flow direction was not always clear from the two-
dimensional network alone. This affected only the direction of the mapped feature, not its position.

## 2.5 Validation

To assess the accuracy of the surface mapping, the patterns of ridges and valleys were compared to AGAP RES data (Corr et
al., 2020) and the BedMachine topography (Morlighem et al; 2020).

---

[1] Before thresholding: fill missing data with zeros*, 5x5 median filter, convert negative values to 0*. Between thresholding
and edge detection: remove areas less than 1 km2, disk-shaped morphological filter, remove missing data cells*. * = step only
used for REMA curvature (necessary due to data gaps).





### 2.5.1 RES profiles

The processed AGAP RES data (Corr et al., 2020) were downloaded as point features, and points constituting twelve segments
of AGAP flightlines that traversed significant ice surface features within the mapped area were arbitrarily selected (Fig. 5).
The bed elevation values from these points were extracted and plotted as two-dimensional profiles. The locations of the mapped
ridges and valleys from both automated and manual methods that intersected these profiles were identified in a GIS and added
to the plots. These plots were inspected on a case-by-case basis to identify ridges and valleys from the bed elevation profiles
and compare them to the locations predicted by each method of surface mapping. Metrics calculated included the proportion
of features successfully matched, the proportion of unsuccessful matches (either where a bed feature was not captured in the
mapping, or where a mapped feature was not observed on the bed), and the mean offset distance between successfully matched
bed features and their mapped locations.

### 2.5.2 BedMachine DEM

The BedMachine Antarctica version 2 bed DEM (Morlighem et al., 2020), which is primarily derived from the AGAP RES
data in this area, was used to compare the planform ridge and valley networks derived via manual mapping to existing
knowledge of the planform landscape structure. Since no bed elevation data were used during the mapping process, this was
an independent test of mapping accuracy, as well as an opportunity to assess whether surface mapping offered any
improvement to the level of detail in the observable planform geometry. Qualitative comparisons were made particularly of
the level of detail available at scales smaller than those generally resolved in the DEM.

## 2.6 Morphometric analyses

In order to understand the structure of the landscape, a series of morphometric parameters were calculated using the manually
mapped ridge and valley networks, as these proved to be the best mapping outputs during the comparisons detailed above.
The lengths of both valley and ridge features were calculated, binned, and plotted as frequency distributions. The orientations
(non-directional) of lines were calculated on a segment-by-segment basis, binned, and plotted as circular histograms (rose
diagrams), with the bin totals scaled according to the total length of line rather than the number of line segments. "Low
confidence" features were included in these calculations, however in the case of ridge features, a second set of histograms was
also plotted, containing the lengths and orientations of the "low confidence" features only, to see whether there was any
difference in directional trend to the overall network. Additionally, for each of these three groups of lines, the angle between
each line segment and the local mean ice flow direction, was calculated and plotted between 0° (line direction parallel to ice
flow) and 90° (line at right angles to ice flow), to assess whether there was a significant relationship between the direction of
ice flow and the orientations of the features that could be observed on the ice surface.
Feature spacing statistics were calculated based on the minimum distance of each cell from the nearest valley or ridge line. In
each case, this statistic was calculated on a 500-by-500 m grid concordant with BedMachine, for the sake of comparability. A





local mean was taken using a circular moving window of radius 25 cells (12.5 km) to produce a more meaningful visualisation,
from which variation in feature spacing across different regions could be assessed. Since the statistic calculated in the first step
is equivalent to half the ridge-to-ridge or valley-to-valley distance, the final statistic was multiplied by two to represent the full
wavelength of feature spacing.

## 3 Results

### 3.1 Automated mapping

Maps resulting from the automated procedure successfully identify linear contrasts in both datasets, and are able to some extent
to categorise them into those that represent valleys and those that represent ridges. There is, however, spatial variation across
both datasets in how comprehensively these two tasks are achieved, which makes visual interpretation more difficult, and
prevents their use in detailed analysis of the valley and ridge networks without further processing.

### 3.2 Manual mapping

The manually-digitised map (Fig. 5) reveals a dendritic valley network, centred around two long central valleys, running
roughly west-to-east. Under ice free conditions, these valleys would form the principal drainage arteries of the mapped area
(Rose et al., 2013). Where they converge, at roughly 81° S, 85° E, they have a combined upstream area of approximately
68,700 km$^2$, more than the next largest drainage unit by an order of magnitude. Several other notably long, straight valleys
within this region have orientations roughly southwest-to-northeast. Outside of the central basin, valleys appear to radiate
outwards towards the lower elevation peripheries of the Gamburtsev Subglacial Mountains, including the parts of the mapped
area outside the densely sampled central grid of the AGAP RES survey (Bell et al., 2011; Ferraccioli et al., 2011).



**Figure 5 – Mapped ridge and valley networks in the study area, overlaid on BedMachine Antarctica bed elevation (Morlighem et al., 2020). Radio echo sounding (RES) profile locations correspond to Fig. 6.**

## 3.3 Validation

The level of detail revealed by manual mapping is a significant improvement on the previous best knowledge of the planform geometry of this area, as seen in the BedMachine DEM, particularly in upland areas, where the frequency of ridges and valleys is below the 5-km minimum line spacing of the AGAP survey grid (Fig. 5). Since the areas of highest bed elevation closely correspond with where ice is thinnest, it is worth noting that the mapping of smaller features in these areas when compared with areas beneath thicker ice may be due to the damping effect of thicker ice on disturbances in flow caused by bed undulations, or by differences in flow caused by differences in basal conditions. However, given the assumption that the valley network does represent what was once a fluvial drainage system, it seems probable that there is to some extent a real change in the frequency of valleys and ridges with elevation, as would be expected from a fluvially-incised landscape.





A total of 139 bed features were identified from 12 RES profiles (Fig. 6), comprising 70 valleys and 69 ridges; 79.9 % of these

features were successfully identified by at least one method, either manual or automated, with 64.8 % matched to a manually

mapped feature, and 51.1 % being matched in one or both automated maps (Fig. 7a). This demonstrates the advantage of using

multiple datasets in the mapping process, and may suggest that in future, a hybrid mapping approach, involving both automated

and manual interpretive steps offers the most comprehensive method of identifying bed features from ice surface data. Among

the manually mapped features along the sampled profiles, 17.8 % were not matched with the indicated type of bed feature

(Table 1). Compared with the figures for the automated maps, (29.3 % for RAMP and 23.7 % for REMA), this demonstrates

the advantage of manual mapping when it comes to avoiding erroneously mapping surface features that are not produced by

topographic disruptions to flow, or that are in fact artefacts in the mapping datasets. Manual mapping also resulted in the

smallest mean horizontal offset for successfully matched features (570 m), with over 90 % of offsets no more than 1 km (Fig.

7b).


**Figure 6 – Bed profiles sampled from AGAP RES survey (Bell et al., 2011; Ferraccioli et al., 2011; Corr et al., 2020), with the locations of mapped ridges (red) and valleys (blue). For profile locations see Fig. 5.**





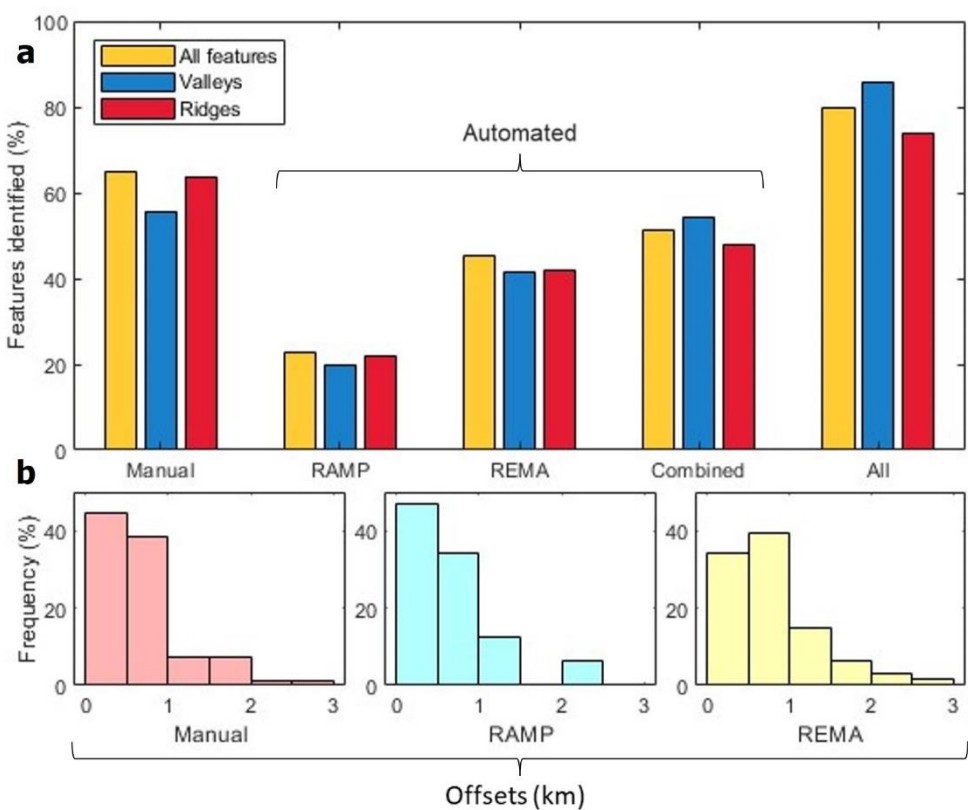

**Figure 7 – Metrics of mapping accuracy, based on twelve sample AGAP radio echo sounding bed profiles: a) percentage of bed**
**features successfully identified; and b) Frequency distributions of horizontal offsets in mapped locations of bed features.**

**Table 1 – Comparing the effectiveness of manual and automated methods in identifying bed features found in selected radar flightlines from the AGAP survey. The optimal value in each column is highlighted in bold.**

| Method | Features identified (%) | Features misidentified (%) | Mean offset distance (km) |
|---|---|---|---|
| Manual | 64.8 | **17.8** | **0.57** |
| Automated (RAMP) | 22.7 | 29.3 | 0.63 |
| Automated (REMA) | 45.3 | 23.7 | 0.74 |
| Automated combined | 51.1 | 29.7 | 0.70 |
| All combined | **79.9** | 30.2 | 0.61 |

### 3.4 Morphometry

The total set of manually mapped valleys and ridges range in length from less than 1 km to nearly 250 km, however the overall
distribution in both cases is log-normal (Fig. 8a), with 61 % of valleys and 55 % of ridges between 1 and 5 km long, rising to



90 % and 85 % respectively between 1 and 15 km. This is largely due to the abundance of short features in higher elevation areas.

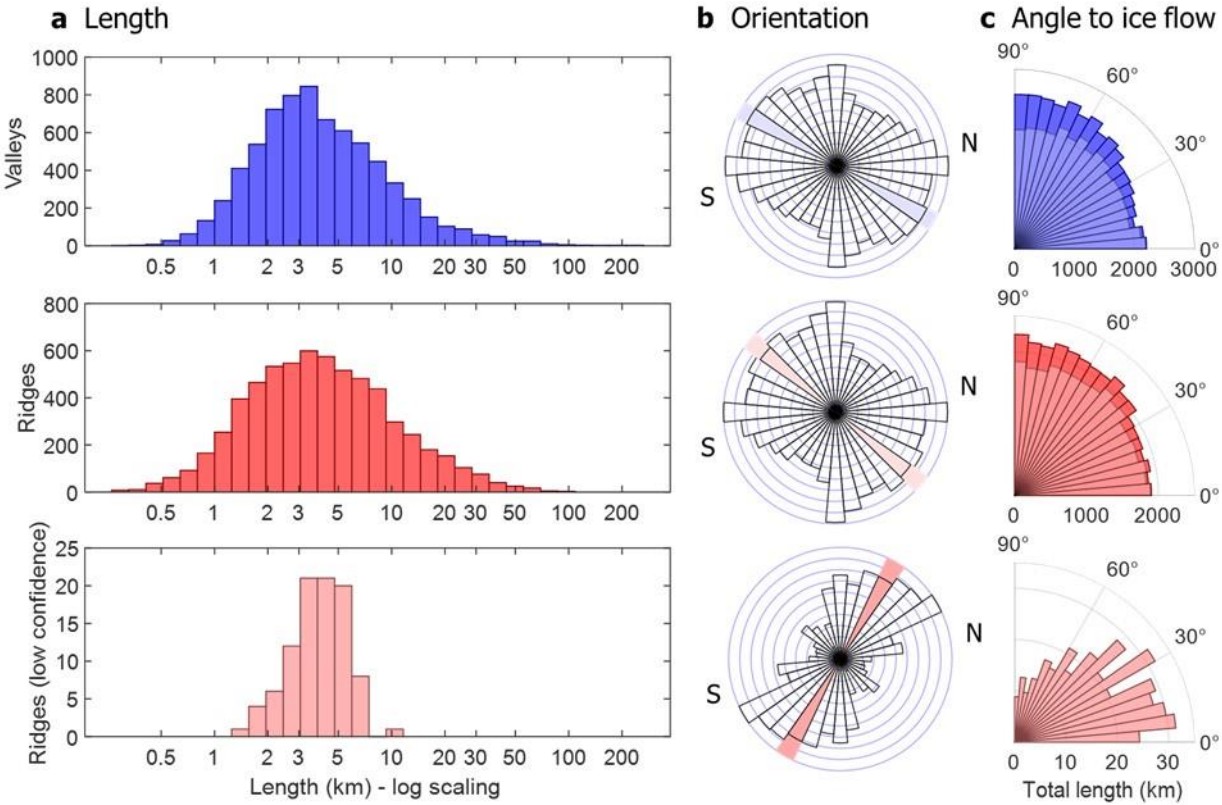

**Figure 8 – a) Length and b) Orientation of mapped valleys (blue), ridges (red), and lower confidence ridges only (pink); c) Angle**
**between mapped valleys/ridges and local mean ice flow direction (derived from Mouginot et al., 2019). Shaded bars in b) denote mean of binned orientations. Horizontal axes in a) are scaled logarithmically, and radial axes in b) and c) record total length of valley/ridge line rather than number of features. Inner bars on the top two plots in c) represent the baseline after removal of the increasing trend from 0° to 90°. Peaks in orientation at top, bottom, left and right of the top two plots in b) are artefacts introduced by the pixelation of the input data.**

The orientations of ridges and valleys are similar, trending on average roughly southwest-to-northeast (Fig. 8b). This orientation is roughly concordant with the Dome A ice divide that lies over the northern ridge of the GSM and roughly perpendicular to the predominant ice flow directions (Fig. 2b). Features oriented perpendicular to flow are more likely to be transmitted to the surface, because they present a greater obstacle to flow than those oriented parallel (Ockenden et al., 2021). Comparison between feature orientations and local ice flow directions (Fig. 8c) indicates significant positive correlations

between angle from flow and total length of valleys ($p = 0.93$) and ridges ($p = 0.92$), with 30 % and 21 % increases in the value of the linear least-squares fit between 0° and 90°, respectively. In both cases, however, the residual is evenly distributed ($|p| < 0.01$), and accounts for 87 % and 90 % of the total length of valleys and ridges, indicating that overall the mapping is not significantly biased by ice flow direction. The "lower confidence" ridges display an opposite trend, with a clear preference for



northwest-to-southeast orientations. The trend in angle to ice flow is also reversed, showing that these ridges are in general
more closely aligned to ice flow. This supports the suggestion that these features exist, and that their absence of expression
from the surface datasets is due to smoother ice flow associated with this alignment of orientation.

Valley widths generally range between 1 and 15 km, however the local mean for 72 % of the mapped area lies between 1.5
and 2.5 km, and is greater than 3 km for less than 2 % (Fig. 9). Valley widths in these upland areas are consistently less than
5 km, which is the minimum spacing of RES survey lines throughout this region. The northern block of the GSM in particular
displays a high density of very narrow valleys, with local means dipping below 1 km in an area that coincides with the mapped
region's highest bed elevations (Fig. 9a).



**Figure 9 – Valley morphometry in the central Gamburtsev Subglacial Mountains: a) Mapped valley network, overlaid on BedMachine Antarctica version 2 bed elevation (Morlighem et al., 2020); b) Local mean valley width, calculated from network in a)**
**as moving-window mean of twice the shortest distance to a mapped valley; c) Frequency distribution of (b).**

## 4 Discussion

### 4.1 Landscape evolution

The mapped valley network bears a predominantly fluvial signature, in the form of a dendritic structure, with identifiable drainage units and a drainage density consistent with that for an alpine environment (Fig. 5). The valley network extends

further than previously detected, maintaining the fluvial signature beyond the central high-elevation region of the GSM into the foothills on the eastern side. This is significant for evincing wider-scale drainage patterns in pre-glacial Antarctica. There is also a greater frequency of small tributary valleys in upland areas than previously mapped, improving the detail of the fluvial network in its source regions. High elevation valley spacing in the central GSM is generally between 1 and 3 km (Fig. 5.7), which is comparable to other alpine mountain ranges such as the Rocky Mountains of North America (Pelletier et al., 2010).

Previous reconstructions of the topography of the GSM (Bo et al., 2009; Rose et al., 2013; Morlighem et al., 2020) have not generally resolved valleys this small, despite features of this size being apparent in RES profiles (Bell et al., 2011). Valley widths may be prone to overestimation in RES profiles, because radar flightlines often intersect valleys obliquely; knowledge of the planform geometry of the valley network is therefore useful in identifying true cross-sections of valley morphology (c.f. Rose et al., 2013).

Unlike other high elevation mountain ranges, the GSM are inferred to be geologically ancient (van de Flierdt et al., 2008; Veevers and Saeed, 2008) and tectonically inactive (Boger, 2011), pre-dating the onset of widespread glaciation in Antarctica at the Eocene-Oligocene transition at 33.7 Ma (Miller et al., 2005; Scher et al., 2011). Given that the GSM are very unlikely to have been deglaciated since this time (Jamieson et al., 2010), it is reasonable to assume that the mapped fluvial network also predates 33.7 Ma (Rose et al., 2013; Paxman et al., 2016), having subsequently been preserved beneath continually cold-

based, non-erosive ice (Van Liefferinge and Pattyn, 2013). The maximum age calculated for this landscape of 230 Ma (Paxman et al., 2016), based on minimal erosion rates (Cox et al., 2010) implies uplift of the GSM much more recently than is otherwise suggested by detrital thermochronology of Prydz Bay marine sediments (van de Flierdt et al., 2008; Veevers and Saeed, 2008; Gupta et al., 2022). These sediments are presumed to have been sourced from the northern GSM via the Lambert Graben; however, the valley network (Fig. 5) indicates that although some sediment would travel by that route, it would not have been

the predominant route for sediments eroded from the central and southern GSM, with the large central basin draining east towards the Ross Sea through what is now the Wilkes Subglacial Basin and/or the Transantarctic Mountains. Moreover, there is evidence of heterogeneity in the landscape structure between the northern block and the rest of the GSM, with valleys more closely spaced in the north (Fig. 9). This potentially suggests a control exerted by differences in the underlying geology of the two regions, hence the Prydz Bay sediments may not be fully representative of bedrock characteristics across the whole GSM.

An alternative possibility is that protective cold-based glaciation was established in the GSM much earlier than 34 Ma (Stoll and Schrag, 1996; Miller et al., 2008), a hypothesis that is consistent with low long-term erosion rates (Cox et al., 2010). It has



previously been suggested that tropical conditions on Antarctica's coast, as evidenced by the occurrence of coal beds (Holdgate et al., 2005; Turner and Padley, 1991), must preclude glaciation of the continent at this time. There is, however, supporting evidence for seasonal sea ice during the Late Cretaceous (Bowman et al., 2013), suggesting conditions also favourable for

terrestrial glaciation. It may be that, with sufficiently strong moisture transport inland, the combination of unusual continentality, high altitude and high latitude that exists in the GSM, allowed for glacial cover of the region even during Greenhouse climates (Miller et al., 2008; Cox et al., 2010). In this case, the fluvial network may be much older, potentially dating back to the last known period of mountain building in East Antarctica, ca. 550–500 Ma (An et al., 2015).

While the planform geometry of the landscape mapped in this study is primarily indicative of fluvial processes, it does provide

some support for one or more phases of localised, erosive glaciation, prior to ice sheet initiation, that modified the existing fluvial landscape (c.f. Bo et al., 2009; Rose et al., 2013). Ridges mapped as "lower confidence" (LC), due to their apparent absence from the mapping data despite logical necessity for a coherent fluvial geometry, may represent locations where ridges have been locally removed by glacial erosion (i.e. glacial breaches; Dury, 1957). A few minor features of this type were identified by Rose et al. (2013), predominantly at high elevations along the major mountain ridgelines of the GSM.

Alternatively, they may persist beneath the ice, but because of their alignment to the local direction of ice flow (Fig. 8), present insufficient obstacle to cause an expression visible on the ice surface (c.f. Ockenden et al., 2021). Since there is diversity in the orientations of LC features, it may be reasonably supposed that those that lie at greater angles to the direction of ice surface flow are more likely to represent glacial breaches.

## 4.2 Basal thermal regime

As discussed, the wholesale preservation of the mapped fluvial valley network (Fig. 5) is significant evidence in favour of consistently cold-based conditions in the GSM throughout their occupation by the East Antarctic Ice Sheet (EAIS; Jamieson et al, 2010; Rose et al., 2013). Several authors therefore suggest that the GSM may host some of the oldest undisturbed basal ice in Antarctica (Fischer et al., 2013; Van Liefferinge and Pattyn, 2013; Wolovick et al., 2021), making them a promising target for drilling "Oldest Ice" cores. Such endeavours include the NSF-funded Center for Oldest Ice Exploration (COLDEX),

which began surveying in the southern GSM during the 2022–23 Antarctic field season. However, basal melting has been seen to occur on small scales (Wolovick et al., 2013) thanks in part to the effects of local topography concentrating geothermal heat flows into topographic lows (Wolovick et al., 2021). Critically, modelling suggests that ridge-valley relief on wavelengths smaller than the spacing of existing RES flightlines (~ 5 km in the GSM) may be enough to induce melting in small-scale upland valleys otherwise assumed to lie within areas of permanently cold-based ice (Wolovick et al., 2021). This presents a

difficult problem when searching for ice drilling sites, as currently available data are not sufficient to accurately identify (and avoid) these localised patches of warm-based ice. Given that the detailed planform geometry presented here is not limited by the spacing or coverage of RES data, and that it records topographic variability on wavelengths consistently below 5 km (Fig. 9), it may therefore be of use in making the predictions of basal thermal conditions required for selecting potential Oldest Ice drilling sites.



### 4.3 Subglacial hydrology

Subglacial topography is also a control on ice sheet hydrology (Wright et al., 2012; Jamieson et al., 2016). In the GSM, there is restricted movement of subglacial water along corridors of low hydraulic potential, defined primarily by the high-relief bed topography: while the subglacial hydrological gradient, dictated by ice surface slope, drives the direction of these flows, the routes they take are determined by the existing subglacial valley network (Wolovick et al., 2013). The mapped valley network may therefore be an indicator of the pathways available to subglacial water beneath the modern ice sheet. The subglacial hydrological pathways of the GSM are predominantly short (Wolovick et al., 2013), and often end in zones of basal refreezing where water is forced up reverse bed slopes under pressure and cools (Bell et al., 2011; Creyts et al., 2014). This occurs on valley sides (where bed topography is transverse to ice surface slope) and in valley heads (where bed topography is aligned with, but opposite to surface slope, such that subglacial water flows up-valley). In the third case, where bed topography is aligned with, and sub-parallel to ice surface slope, longer-distance transport may be possible, uninterrupted by basal refreezing. The major valleys that would have drained the palaeo-GSM to the east (Fig. 5) are in relative accordance with the direction of ice surface slope, potentially offering a subglacial hydrological connection from the GSM to the wider EAIS. This may be complicated by the transport of water up reverse bed slopes created by overdeepenings along valley long-profiles, which are known in the deep valleys of the GSM (Bo et al., 2009; Rose et al., 2013), but not possible to infer from planform geometry alone.

### 4.4 Ice sheet evolution

The preservation of the mapped valley network beyond the central region of the GSM is significant, because it demonstrates that, as inferred in the central GSM (Bo et al., 2009; Rose et al., 2013), glaciation of the surrounding landscape has been minimally erosive, or at least, that the pattern of ice flow, and hence the pattern of erosion, has been predominantly guided by the pre-existing fluvial geometry. This is consistent with the idea that the GSM were a key source of ice during the early Oligocene expansion of glaciation in Antarctica (DeConto and Pollard, 2003), because the marginal regions of an expanding ice cap centred on the GSM would have been initially thin and topographically confined, like many modern-day Arctic ice-caps which have topographically confined outlets (e.g. Baffin Island ice cap, Svalbard, Severnaya Zemlya and margins of SE Greenland). As these ice margins grew, erosion would have been concentrated along these topographic lows (Sugden and John, 1976), leading to a feedback whereby continued erosion promoted increased flow of ice along the same topographic corridors, further focusing erosive power along these routes (Jamieson et al., 2008, Kessler et al., 2008). Even where ice has not remained exclusively cold-based, therefore, the broad strokes of the fluvial valley network of pre-glacial East Antarctica may be more widely preserved than previously understood, indicating remarkable stability at the core of the EAIS. The accordance between the positions and orientations of the central ridge of the GSM and the overlying Dome A ice divide (Fig. 2, 5) supports the idea that this spine of high topography may have remained a keystone for the EAIS, preventing the deglaciation of one of its major source regions during climate-induced oscillations (Wolovick et al., 2021).



## 5 Conclusions

In this study, a new map of the planform ridge and valley geometry of the central Gamburtsev Subglacial Mountains (GSM), East Antarctica, was produced by using satellite remote sensing data to identify and interpret changes in ice surface slope.
Manual and automated approaches to processing these data were tested, and existing bed elevation datasets were used to validate the correspondence between the resulting maps and the known bed topography. Furthermore, the morphometry of the manually-mapped networks was analysed, revealing details about the structure of the pre-glacial fluvial landscape and its subsequent evolution under early phases of Antarctic glaciation. Implications of this increased knowledge of the basal topography for the evolution of the East Antarctic Ice Sheet, preservation of ancient ice, and its subglacial hydrological systems
were also discussed. Key findings of the work were as follows:

1.    Mapping of subglacial topography from ice surface curvature is validated for the GSM by existing measurements of ice thickness from radio echo sounding (RES), and an existing model of bed topography produced using the RES data. The maps presented here expand both the coverage and the level of detail available for the planform geometry of the GSM, particularly in high bed elevation/thinner ice areas.

2.    Manual mapping identified a greater proportion of bed features (64.8 %) than automated mapping (51.1 %), produced fewer erroneous identifications, and had smaller offsets between mapped and actual features. The proportion of features successfully identified was increased further (79.9 %) when all methods were considered together, suggesting that future mapping would maximise accuracy and comprehensiveness by combining automated and manual approaches.

3.    The mapped valley network preserves information about the pre-glacial fluvial regime, suggesting the former
existence of a large central catchment (68,700 km$^2$) draining east towards the Ross Sea.

4.    There is some evidence for modification of the fluvial valley network by local- to regional-scale erosive ice through the breaching of fluvial drainage divides and the overdeepening of valley long-profiles, however, uncertainty in the mapping of these features produces ambiguity in their interpretation.

5.    Care must be taken when interpreting the data presented here to account for the limitations inherent in mapping bed
features from the surface. Orientations of features represented may be biased by the direction of ice flow, with features aligned to flow less likely to produce a surface expression. The detail available also varies with the thickness of the ice column, with thicker ice dampening the effects of subglacial topography on flow. This can lead to ambiguity over the presence or interpretation of mapped features, suggesting the importance of using this approach in conjunction with other methods of mapping ice sheet beds, such as radio echo sounding.

6.    The preservation of the pre-glacial fluvial valley network more widely than previously known indicates, for the area surrounding the GSM, long-term ice sheet behaviour which is either consistently non-erosive, or sufficiently influenced by the bed topography to concentrate erosion in pre-existing topographic lows. This is suggested to document the importance of the GSM as a centre of growth for the early East Antarctic Ice Sheet, and as a stabilising influence during its subsequent evolution.



7.      The minimum wavelength of detectable topographic variability (< 5 km) is smaller than can be reproduced in bed
models using existing data. Ridge and valley structures on this scale may play an important role in governing local fluctuations
in bed conditions, including occurrences of basal melting and routing of subglacial water flows. Maps of planform landscape
geometry such as those presented here may therefore be useful in evaluating sites for the preservation of Oldest Ice (> 1 Myr)
cores in regions of highly variable subglacial topography.

8.      In addition to the applications mentioned, the production of synthesised bed elevation models using the mapped
network, or through combining the mapped network with existing data, has not been attempted, but may be an avenue to
explore in future. Such products could be of use for ice sheet models that seek to simulate more accurately the effects of high-
relief basal topography on ice flow or basal hydrology.

**Data and code availability**

Bedmap2 bed elevation and errors are available for download at https://data.bas.ac.uk/full-
record.php?id=GB/NERC/BAS/PDC/01617 (Fretwell et al., 2013). BedMachine Antarctica bed elevation is available for
download at https://nsidc.org/data/nsidc-0756/versions/3 (Morlighem et al., 2020). Reference Elevation Model of Antarctica
products are available for download at https://www.pgc.umn.edu/data/rema/ (Howat et al., 2019). The RADARSAT-1
Antarctic Mapping Project image mosaic is available for download at: https://doi.org/10.5067/8AF4ZRPULS4H (Jezek et al.
2013). The Antarctica's Gamburtsev Province radio echo sounding data are available for download at
https://doi.org/10.5285/0F6F5A45-D8AF-4511-A264-B0B35EE34AF6 (Corr et al., 2020).

The code for automated processing of the REMA and RAMP datasets will be provided by Edmund J. Lea upon request.

**Author contributions**

EJL managed the project and conducted the mapping, coding, and analyses, under supervision from SSRJ and MJB. EJL
prepared the initial draft, and SSRJ and MJB provided comments and revisions.

**Competing interests**

The authors declare that they have no conflict of interest.

**Acknowledgements**

This work received no funding.



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
