# Peer review of "Alpine topography of the Gamburtsev Subglacial Mountains, Antarctica, mapped from ice sheet surface morphology"

_The Cryosphere, 2023_

## Referee Comment (RC1)

**General comments**

This paper explores a novel method for mapping the geometry of the Gambertsev Subglacial Mountains using ice surface roughness and slope. Automatic and manual mapping methods are combined to produce detailed maps of ridge and valley morphology for areas between existing radar flight-lines, expanding the pre-glacial fluvial valley network beyond what was previously known. The authors conclude that this map could be of use for selecting drill sites for the oldest ice, or for ice sheet models studying the effect of basal topography on ice flow or hydrology.

The paper is generally very well written and structured. Before publication though, I would like to see an expansion of the methodology for the automated mapping, which I do not think is currently sufficiently detailed to be reproducible by other scientists, but could easily be made to be. The code for the paper is also currently only available on request, and I would like to encourage the authors to make this openly accessible (e.g. through GitHub or Zenodo), which would help with reproducibility. Otherwise, I have only a few minor points.

**Specific comments**

141 Would you recommend a neighbourhood distance of 1000m if applying this method elsewhere? It would be interesting to see some discussion as to whether you think this value would vary with the ice thickness, or perhaps a trade-off plot between noisiness and crispness of surface features to give a clearer idea of how you came to your decision.

164 Up to this point, I think that your methodology is very clear, and it is well explained which metrics you are focussing on and how those are calculated. However, I would like to see this section expanded further to make this method more reproducible. How was the adaptive binary threshold calculated? Which algorithm did you use for edge-detection? You say that the code for the paper is available on request, but is that the code for this step? I think that the pre and post processing steps should also be explained in the main text, rather than as a footnote.

235 Could you present a figure of the features identified with automated mapping? It would be nice to see a visual representation of the features which are found with the manual mapping but not the automated (and vice versa), especially since you mention that you do the manual mapping in a deliberately interpretive way to try and fill in some gaps. Some kind of visual comparison would be good, and would give a better idea of the overall coverage of features, since the later comparisons in section 3.3 (as far as I can tell) are only to the AGAP radar data?

358 Modelling suggests that for ice sliding over the bed, bed topography is transferred to the ice surface. However, earlier in the paper you mention that the ice is most likely frozen to the bed, allowing it to preserve the pre-glacial topography. Does that mean that the ice is not sliding over the bed, and flowing slowly through internal deformation, and do you think that would have an influence on the way in which bed topography is transferred to the ice surface in this region?

**Technical comments**

79 I feel like this could be rephrased with fewer commas to read more smoothly. 'The geology of the Gamburtsev Subglacial Mountains is poorly understood, with their…'

100 You could be more explicit about the directionality of the relationship between oxygen isotopes and glaciation here; 'as increased terrestrial ice volume leads to a higher d18O in seawater and hence in benthic organisms.'

115 It is very difficult to see the surface curvature in Figure 2d which is overlain on the RAMP intensity. I would suggest plotting these two separately, or increasing the transparency of the RAMP Intensity layer.

124 Bed elevation models? Or a bed elevation model?

145 Figure 3, could you include a colour scale bar for mean curvature with this figure. If the scale is different for each plot, perhaps a normalised colour scale could be used.

169 REMAv2 has many fewer gaps than REMAv1, so I'm curious to know which version was used (there is no timestamp on the reference for the dataset). Do you think that these gaps made a significant impact to the results of your study?

209 It might be good to mention that the technique used by Bedmachine Antarctica to go from the AGAP RES data to their DEM is streamline diffusion (and not mass conservation as many readers might assume).

---

## Author Comment (AC1)

**Overview**

*We would like to thank both reviewers and the editor for their positive feedback and constructive comments, and acknowledge an additional comment received via email from a member of the community who has experience researching the subglacial geology of East Antarctica. We respond here to all of these comments and propose what we hope are appropriate changes to the submitted manuscript. Please note that this document contains responses to all reviewers' and additional comments received.*

**Reviewer #1 comments**

This paper explores a novel method for mapping the geometry of the Gambertsev Subglacial Mountains using ice surface roughness and slope. Automatic and manual mapping methods are combined to produce detailed maps of ridge and valley morphology for areas between existing radar flight-lines, expanding the pre-glacial fluvial valley network beyond what was previously known. The authors conclude that this map could be of use for selecting drill sites for the oldest ice, or for ice sheet models studying the effect of basal topography on ice flow or hydrology.

The paper is generally very well written and structured. Before publication though, I would like to see an expansion of the methodology for the automated mapping, which I do not think is currently sufficiently detailed to be reproducible by other scientists, but could easily be made to be. The code for the paper is also currently only available on request, and I would like to encourage the authors to make this openly accessible (e.g. through GitHub or Zenodo), which would help with reproducibility. Otherwise, I have only a few minor points.

*We thank the reviewer for their positive comments on the paper and agree with the suggestion that the code used in the paper be made openly accessible. Much of the improved methodological detail requested exists in the form of an unpublished (but accessible online) Master's thesis by the lead author. The current balance of detail was aimed to ensure the paper was accessible and the main findings of the research were emphasised rather than providing a technical overview. We propose that we will include a more detailed description of the method (sufficient to allow reproducibility)  as a supplement to preserve an appropriate balance of detail in the main body of the paper.*

*We include for reference a figure from the thesis mentioned above which outlines the automated mapping procedure step by step and we would include this in the supplement with the description. We believe that in conjunction with the original code being made publicly accessible, such an outline of the method would be sufficient to allow it to be reproduced by other scientists. Further detail is provided in answers to specific comments below where we also update some of the main text and figures to aid understanding of the robustness of the methods.*

[Figure]

*Figure to be adapted for inclusion in supplement – Summary of automated mapping procedure. Symbology will be described fully. Steps marked with \* were necessary only when using REMA curvature as the input surface dataset (due to data gaps).*

**Specific comments**

141 Would you recommend a neighbourhood distance of 1000m if applying this method elsewhere? It would be interesting to see some discussion as to whether you think this value would vary with the ice thickness, or perhaps a trade-off plot between noisiness and crispness of surface features to give a clearer idea of how you came to your decision.

*We found that the 1000 m neighbourhood distance seemed widely applicable throughout the study area, bearing in mind that levels of noise in the dataset vary significantly. We will add a sentence that reflects on the wider applicability of this value.*

164 Up to this point, I think that your methodology is very clear, and it is well explained which metrics you are focussing on and how those are calculated. However, I would like to see this section expanded further to make this method more reproducible. How was the adaptive binary threshold calculated? Which algorithm did you use for edge-detection? You say that the code for the paper is available on request, but is that the code for this step? I think that the pre and post processing steps should also be explained in the main text, rather than as a footnote.

*We propose to include the requested information as a supplement (see previous comment above) to allow the method to be reproduced more easily, as per our initial comment. All the code for the whole automated mapping process will also be made openly available via a repository such as Zenodo.*

235 Could you present a figure of the features identified with automated mapping? It would be nice to see a visual representation of the features which are found with the manual mapping but not the automated (and vice versa), especially since you mention that you do the manual mapping in a deliberately interpretive way to try and fill in some gaps. Some kind of visual comparison would be good, and would give a better idea of the overall coverage of features, since the later comparisons in section 3.3 (as far as I can tell) are only to the AGAP radar data?

*We propose including a figure in section 3.1 showing extracts from the automated mapping which demonstrate the differences between the automated methods and the manual mapping.*

*Regarding the AGAP radar comparisons in section 3.3, we confirm that the comparisons were made for both automated maps as well as the manual map, with summary metrics presented in Figure 7 and Table 1. Only the manual comparison is shown visually in Figure 6 as a demonstration of the approach – please note that this section will also be updated following comments from Reviewer 2 to reflect a more robust method for identifying the bed ridges and valleys from the radar data and matching them to mapped features.*

358 Modelling suggests that for ice sliding over the bed, bed topography is transferred to the ice surface. However, earlier in the paper you mention that the ice is most likely frozen to the bed, allowing it to preserve the pre-glacial topography. Does that mean that the ice is not sliding over the bed, and flowing slowly through internal deformation, and do you think

that would have an influence on the way in which bed topography is transferred to the ice surface in this region?

*Yes, models consistently indicate that ice is predominantly frozen and not sliding (DeConto and Pollard, 2003; Creyts et al. 2014; Wolovick et al. 2021) – this does suggest that the mechanism for transferring the topographic signature to the ice surface must be different to the process observed in sliding ice. However the results indicate that ice moving predominantly through internal deformation is still affected by bed topography, especially when the bed relief is significant (c.f. Ross et al., 2014). Flow of ice under these conditions can be extremely complicated (Martinez, 2021) and modelling it is beyond the scope of this paper. Further understanding the processes involved would be an intriguing goal for future work. Nonetheless, the results indicate that certain findings of Ockenden (2021) are still pertinent, including the underrepresentation of landforms that are oriented parallel to modern ice flow in the ice surface morphology. We will make it clear in the text that, while the overriding principle is similar, the processes involved in the two different contexts may be different.*

**Technical comments**

79 I feel like this could be rephrased with fewer commas to read more smoothly. 'The geology of the Gamburtsev Subglacial Mountains is poorly understood, with their…'

*We will rephrase the sentence accordingly.*

100 You could be more explicit about the directionality of the relationship between oxygen isotopes and glaciation here; 'as increased terrestrial ice volume leads to a higher d18O in seawater and hence in benthic organisms.'

*We will include a short explanation of the link between ice volume and d18O as suggested.*

115 It is very difficult to see the surface curvature in Figure 2d which is overlain on the RAMP intensity. I would suggest plotting these two separately, or increasing the transparency of the RAMP Intensity layer.

*This is an important observation for which we are grateful. Figure 2 will be adapted to have an additional panel so that the RAMP intensity and REMA curvature datasets can each be shown separately.*

124 Bed elevation models? Or a bed elevation model?

*We propose rephrasing this sentence to be specific about the datasets used: "… and assessed the results against the AGAP radar survey bed elevation data (Bell et al. 2011; Ferraccioli et al., 2011), and the BedMachine bed elevation model (Morlighem et al., 2020)."*

145 Figure 3, could you include a colour scale bar for mean curvature with this figure. If the scale is different for each plot, perhaps a normalised colour scale could be used.

*We will produce a version of this figure with a normalised scale and scale bar as suggested.*

169 REMAv2 has many fewer gaps than REMAv1, so I'm curious to know which version was used (there is no timestamp on the reference for the dataset). Do you think that these gaps made a significant impact to the results of your study?

*We will indicate in the text that REMAv1 was used (as REMAv2 did not exist when the mapping was conducted). We will also include a statement of the data coverage for the study area to indicate that missing data had a minimal impact on the results (Data coverage over the study area is very good, with data in 99.92% of 200-by-200 m cells).*

209 It might be good to mention that the technique used by Bedmachine Antarctica to go from the AGAP RES data to their DEM is streamline diffusion (and not mass conservation as many readers might assume).

*We will include this information as suggested e.g. "The BedMachine Antarctica version 2 bed DEM (Morlighem et al., 2020), which is primarily derived from the AGAP RES data in this area using the streamline diffusion technique (Morlighem et al., 2010), …"*

**Reviewer #2 Comments**

Lea et al present a map of ridge and valley features based on satellite observations of the surface. They validate the inferences using the dense AGAP radar survey. This technique allows mapping beyond the radar survey, and, in theory, resolution at length scales shorter than the radar line spacing. The technique is clever and offers additional information to the radar data. The paper would be much improved from a more quantitative validation with objective definitions of ridges and valleys in the radar data as the qualitative assessment (i.e. Figure 6) is not particularly convincing. The technique, with additional quantitative validation, will be a useful addition to tools for inferring subglacial features in regions undersampled by radar data.

*We thank the reviewer for their positive comments on the technique developed in the paper, and acknowledge the need for a more robust procedure to quantitatively validate it with regards to objectively defined bed ridges and valleys. We address these concerns through an updated version of the procedure outlined in response to specific comments below.*

The technique is, in general, well described. It would be useful to expand Figure 4 to include both a valley as well as short length scale identifications of valleys and ridges, e.g. segment 10 10-16km. Figure 4 is a good example, but likely lacks many of the complications that arise between manual and automated interpretation.

*It is true that Figure 4 does not cover every eventuality when it comes to mapping by hand, as it is intended to be illustrative of the technique only. Following comments from Reviewer 1 also, we propose including a more detailed description of both the manual and automated methods as a supplement to the main paper to address this and similar concerns.*

The validation suffers from a lack of quantification of valley and ridges in the radar data. How a ridge and a valley are defined in the radar data is not clear. The authors provide statistics of the percentage of features that are correctly identified, but do not describe how the "true" features are defined. Figure 6 shows the inferred features, and the bed data, but

not the "true" ridges and troughs defined by the radar data. Without a specific definition of ridges and troughs, quantitative metrics of how many features were correctly and incorrectly identified do not convince me. In Figure 6, segment 10, 5 features are identified between 10 and 16km, but the features are very small in scale and more significant features between 3 and 10 km are not identified.

Figure 6 also reveals that the same scales of basal features are not consistently identified between segments. For instance, small features are identified in segment 10, but are not identified in Figure 9. The bed elevation is similar between them, so it's unlikely that a greater ice thickness is obscuring the bed variations.

*We will implement a modified validation procedure which identifies ridges and valleys as local maxima/minima in the radar data with minimum 100 m prominence. This excludes some 40 "true" features originally identified qualitatively, reducing the sample size used in validation, but as the Reviewer notes, it makes the quantitative metrics more robust and meaningful. Figures 6 and 7 and Table 1 will be updated accordingly. We also propose additional discussion of the spatial variability in scale of features identified similar to the following:*

*"The variable ability of the method to pick up smaller-scale features is a result of the spatial variability in the degree to which the topography is transferred to the surface (c.f. Chang et al., 2016), which depends not only on ice thickness, but also direction of ice flow (c.f. Ockenden et al., 2021), and other factors such as distance from the ice divide e.g. Figure 2d shows how RAMP intensity has greater contrast further from the ice divide, reflecting overall steeper ice surface slopes further from the Dome A summit. As many features were digitised as possible in each region, but the maximal "resolution" of the manual method thus varies spatially. For this reason, a range of different flightline sections from across the study area were chosen as the basis for the validation, to illustrate these discrepancies [such as those identified by the Reviewer in Figure 6]."*

It would be much more useful to have the "true" ridges and troughs marked, solid lines that indicate what the authors consider a successfully identified feature, and dashed lines for incorrect features. Further, the authors need to define a distance that the mapped feature must match the true feature – or possibly define a ratio of distance to amplitude of feature.

*An updated version of Figure 6 (see next page) following the modified validation procedure adopts these suggestions - "true" bed features are marked using red and blue triangles, matched features using solid lines, unmatched features as dashed lines (offsets labelled). The distance cut-off for matching features was set at 2 km (as very few matched features were more than 2 km distant from the "true" feature in the original validation procedure).*

It is surprising to me how often the same features are identified successively (i.e. 2 or more ridges in a row). For instance, Figure 6, segment 9 has 3 ridges between 5 and 10 km without any valleys in between. And then the only valley is identified at a ridge at just over 15km. It seems like a sequence of multiple of the same feature in a row would be a good check that could be implemented into the automated routine to help improve its performance.

*This occurs mainly as a function of sampling the two-dimensional planform network along the discrete flightlines traced by the radar data. Each vertical line in Figure 6 represents a point where the flightline intersects a mapped ridge or valley. Because ridges and valleys are generally only digitised as far as they are visible in the datasets used for mapping (RAMP, REMA curvature), there is no guarantee of a strict ridge-valley-ridge progression along a given flightline. This is the case, for instance, where flightlines cross the upper portions of valleys (where the valley may not have been digitised all the way to the valley head) or broader valleys close to confluences (where there may not be a discernible ridge between channels). A more detailed explanation of the updated validation procedure will be included to make this sampling relationship more explicit.*

*The automated routine (as will be detailed in the proposed supplement) evaluates on a pixel-by-pixel basis whether a pixel is identified as a ridge or valley by its relationship to its immediate neighbours, and does not intrinsically consider proximity to other features in any given direction. We agree that such a check would be beneficial in a more advanced version of the procedure, but note that it is not straightforward to implement in two dimensions and so will not be included here. We will also include a sentence in the main text which summarises this procedure.*

*We also note that the instances identified by the Reviewer in Figure 6 are all examples from manual mapping only, and that this will be made clearer in the final version.*

I would also like to see a more quantitative analysis of the acceptable distance between a "true" and a mapped feature. There are many instances in Figure 6 where there are potentially multiple features in the data that the mapped feature could be assigned to.

*The modified validation algorithm assigns a bed feature to the nearest mapped feature that is not closer to a different bed feature (to which it is therefore assigned). No more than one mapped feature is assigned to each bed feature, and no mapped feature is matched if it is more than 2 km distant from the nearest bed feature. Note that this procedure is limited by the one-dimensional nature of the radar data being used for comparison – it is entirely possible that unmatched features in Figure 6 represent ridges or valleys with offsets in a lateral direction (i.e. into or out of the page rather than along the flightline). The variability in offset distance and direction appear to be related to local variability in ice flow, as will be indicated in the final version of Figure 6 which we propose should be updated to the following:*

[Figure]

While most of my comments have to do with the validation, there are a couple of other areas where the paper could be improved. A morphological analysis of a few alpine regions that could be directly compared with GSM would be very informative. How do the Rockies compare quantitatively? Is GSM closer to the northern Rockies (i.e. Glacier NP) or the southern Rockies (ie. The San Juans in CO). Or I'm sure you come up with better analogs to determine the extent of glaciation that may have occurred.

*Assuming this comment refers to the morphometric analyses of the valley network, it is possible to replicate some of the analyses performed by using valley networks derived from DEMs, such as the global dataset constructed by Lin et al. (2021). The primary metric used in the paper that could be compared quantitatively is ridge-to-ridge valley width (valley spacing) – our original intention was mainly to demonstrate using this analysis that valleys in the GSM have widths smaller than those identifiable using other methods, and that these widths are on par with those found in well-known alpine environments. To this end the paper does reference observations by Pelletier et al. (2010) of typical first-order valley widths in three North American ranges (namely, the Beartooth-Absaroka Mountains, Montana; the Uinta Mountains, Utah; the central Rocky Mountains, Colorado). They note that these lie generally in the range 1-3 km, which accords well with the range displayed in Figure 9 of this paper for the GSM. Given that the method applied here is only a proxy for true valley widths, which can be measured directly in other ranges from a surface DEM, it seems unlikely that any more specific comparison could be made based on applying it in these locations. Therefore, in order to avoid reprising analyses that have previously been conducted by other researchers, we propose:*

- *To add detail to the comparison with the observations of Pelletier et al. (2010), perhaps with reference to their Fig. 2, which visually displays the typical valley spacing for the three ranges they investigated.*
- *To include as supplementary details or in an online repository, the GIS workflow used to create the raster of mean valley spacing used in Figure 9, so that it can more easily be applied to other vector representations of valley networks (e.g. Lin et al., 2021)*

*The authors are sceptical that it is feasible to determine the extent of a past glaciation in the GSM using only the planform networks derived in this study, and note that this was never an intended aim of the paper.*

**Additional Comment (received via email)**

I'm writing to let you know I really enjoyed reading your Discussion paper in TC on alpine topography of the GSM. It's very well presented and well written, so congratulations for bringing together a focused new study on the topic.

As you take in outside review input, I have just a couple of suggestions for minor revisions or discussion points:

Fig. 1 — AGAP not shown on figure, which corresponds to unlabeled box with dashed outline?

*The dashed box corresponds to the central mesh of the AGAP Survey. This is indicated in the Figure legend but will be made clearer with a label on the Figure itself.*

Fig. 2b, label ice catchments.

*This is a very good suggestion to aid the discussion of sediment provenance and subglacial pathways later in the paper, and will be implemented.*

I'm attaching a couple of papers that are relevant to your discussion of GSM origin scenarios and might be of interest. Together these address origins of igneous crust in central East Antarctica (Goodge et al., 2017) and its thermal evolution during the Paleozoic to Cenozoic (Fitzgerald and Goodge, 2022). These studies of glacially-eroded clasts may, in contrast to the cited studies from debris in Prydz Bay, give geological clues to the composition and age of the enigmatic GSM. In particular, material sourced from the southern flank of the GSM and transported into the Byrd Glacier drainage have Proterozoic igneous ages and cooling ages ≤500 Ma that together point to a rather ancient origin. To be sure, we don't know how far these cobbles were transported, so their source might not be as far as the GSM, but they seem to provide the most tangible geological evidence for crust in the region, particularly as their igneous ages are largely unknown elsewhere in exposed parts of Antarctica, and they preserve a long cooling and exhumation history.

*Key findings from these papers will be included in the discussion of Gamburtsev age and origin in section 1.1 and/or later in the paper.*

**Editor Comment**

Thank you for submitting "Alpine topography of the Gamburtsev Subglacial Mountains, Antarctica, mapped from ice sheet surface morphology" to The Cryosphere. Two reviews have now been received.

The reviewers comment that the manuscript is well written, well structured, and see the potential for this methodology to be a useful tool for inferring subglacial features once the suggested changes have been incorporated.

Both reviewers have requested improvements in the methods related to automatic mapping to ensure reproducibility (e.g. specifically defining the binary threshold, algorithm for edge detection, and how "true" ridges and valleys were actually defined in the radar data) and improved validation (e.g. additional or expansion of figures to include automatic mapping results and quantitative validation of radar results). In addition, Reviewer 2 requests that you apply the methodology to other mountain ranges. As Editor I ask that you please address each and every one of their comments.

We thank you for your work on this paper and the promptness of your comment following the reception of the two reviewer comments. We believe the amendments and additions detailed above would be sufficient to address their concerns around reproducibility of the method and reliability of the quantitative validation. Regarding Reviewer 2's comment on the use of other mountain ranges as analogues for the mapped landscape, we believe this was intended to suggest comparison of the morphometric analyses (e.g. ridge orientation,

valley spacing etc.) rather than a rerun of the mapping methodology (which is only applicable in the case of ice-covered landscapes).

**Other**

Line 318 – wrong figure reference, should say Fig. 9 at end of line, not Fig. 5.7 – this will be updated.

**References**

Creyts, T.T. et al. (2014) 'Freezing of ridges and water networks preserves the Gamburtsev Subglacial Mountains for millions of years', Geophysical Research Letters, 41(22), pp. 8114–8122. Available at: https://doi.org/10.1002/2014GL061491.

DeConto, R.M. and Pollard, D. (2003) 'Rapid Cenozoic glaciation of Antarctica induced by declining atmospheric CO2', Nature, 421(6920), pp. 245–249. Available at: https://doi.org/10.1038/nature01290.

Fitzgerald, P.G. and Goodge, J.W. (2022) 'Exhumation and tectonic history of inaccessible subglacial interior East Antarctica from thermochronology on glacial erratics', Nature Communications, 13(1), p. 6217. Available at: https://doi.org/10.1038/s41467-022-33791-y.

Goodge, J.W. et al. (2017) 'Proterozoic crustal evolution of central East Antarctica: Age and isotopic evidence from glacial igneous clasts, and links with Australia and Laurentia', Precambrian Research, 299, pp. 151–176. Available at: https://doi.org/10.1016/j.precamres.2017.07.026.

Lin, P. et al. (2021) 'A new vector-based global river network dataset accounting for variable drainage density', Scientific Data, 8(1), p. 28. Available at: https://doi.org/10.1038/s41597-021-00819-9.

Martinez, E. (2021) 'Moffatt Eddies in Subglacial Mountain Valleys', ENGS 88 Honors Thesis (AB Students) [Preprint]. Available at: https://digitalcommons.dartmouth.edu/engs88/28.

Ockenden, H. et al. (2021) 'Inverting ice surface elevation and velocity for bed topography and slipperiness beneath Thwaites Glacier', The Cryosphere Discussions, pp. 1–34. Available at: https://doi.org/10.5194/tc-2021-287.

Pelletier, J.D., Comeau, D. and Kargel, J. (2010) 'Controls of glacial valley spacing on earth and mars', Geomorphology, 116(1), pp. 189–201. Available at: https://doi.org/10.1016/j.geomorph.2009.10.018.

Ross, N. et al. (2014) 'The Ellsworth Subglacial Highlands: Inception and retreat of the West Antarctic Ice Sheet', GSA Bulletin, 126(1–2), pp. 3–15. Available at: https://doi.org/10.1130/B30794.1.

Wolovick, M.J., Moore, J.C. and Zhao, L. (2021) 'Joint Inversion for Surface Accumulation Rate and Geothermal Heat Flow From Ice-Penetrating Radar Observations at Dome A, East Antarctica. Part II: Ice Sheet State and Geophysical Analysis', Journal of Geophysical

Research: Earth Surface, 126(5), p. e2020JF005936. Available at: https://doi.org/10.1029/2020JF005936.

---

## Author Response (AR2)

**Overview**

*This document details the changes made in the revised version of the manuscript, according to the responses made to reviewers' comments previously. We are grateful to the two reviewers and the additional comment received directly for their help in improving the manuscript.*

**Reviewer #1 comments**

The paper is generally very well written and structured. Before publication though, I would like to see an expansion of the methodology for the automated mapping, which I do not think is currently sufficiently detailed to be reproducible by other scientists, but could easily be made to be. The code for the paper is also currently only available on request, and I would like to encourage the authors to make this openly accessible (e.g. through GitHub or Zenodo), which would help with reproducibility. Otherwise, I have only a few minor points.

*We thank the reviewer for their positive comments on the paper and agree with the suggestion that the code used in the paper be made openly accessible – it is now available on Zenodo and the doi is 10.5281/zenodo.10550538. Access is currently restricted but the following link should provide private access:*

*https://zenodo.org/records/10550538?token=eyJhbGciOiJIUzUxMiJ9.eyJpZCI6IjFlMWNlMG M2LTdkYTUtNDBhNi1iYjc3LWQ0YTg5MzNkNmZkYSIsImRhdGEiOnt9LCJyYW5kb20iOiJmMTl mZDU2NWQxNzlhMDM1MDQxNTBhNWYxZTU3ZWYwYSJ9.o28cwjZ4I3MiHK9bOutHP4AJt2k 2j866HwJ7iXYPIbMsGTUyQ6aUpseMb41aQL-mw3oBIHKtQSKFUK62Rwq4GA*

*We include a more detailed description of the method (sufficient to allow reproducibility) in a supplement. Further detail is provided in answers to specific comments below where we also update some of the main text and figures to aid understanding of the robustness of the methods.*

**Specific comments**

141 Would you recommend a neighbourhood distance of 1000m if applying this method elsewhere? It would be interesting to see some discussion as to whether you think this value would vary with the ice thickness, or perhaps a trade-off plot between noisiness and crispness of surface features to give a clearer idea of how you came to your decision.

*We found that the 1000 m neighbourhood distance seemed widely applicable throughout the study area, bearing in mind that levels of noise in the dataset vary significantly. We will add a sentence that reflects on the wider applicability of this value. Annotations added to Fig. 3 indicate factors considered when choosing which version to use.*

164 Up to this point, I think that your methodology is very clear, and it is well explained which metrics you are focussing on and how those are calculated. However, I would like to see this section expanded further to make this method more reproducible. How was the adaptive binary threshold calculated? Which algorithm did you use for edge-detection? You

say that the code for the paper is available on request, but is that the code for this step? I think that the pre and post processing steps should also be explained in the main text, rather than as a footnote.

*We include the requested information as a supplement (see previous comment above) to allow the method to be reproduced more easily. All the code for the whole automated mapping process has been uploaded and given a DOI using Zenodo.*

*We clarify in the text that the edge-detection algorithm is custom-built.*

235 Could you present a figure of the features identified with automated mapping? It would be nice to see a visual representation of the features which are found with the manual mapping but not the automated (and vice versa), especially since you mention that you do the manual mapping in a deliberately interpretive way to try and fill in some gaps. Some kind of visual comparison would be good, and would give a better idea of the overall coverage of features, since the later comparisons in section 3.3 (as far as I can tell) are only to the AGAP radar data?

*We include an additional figure in section 3.1 showing extracts from the automated mapping which demonstrate the differences between the automated methods and the manual mapping.*

*Regarding the AGAP radar comparisons in section 3.3, we confirm that the comparisons were made for both automated maps as well as the manual map, with summary metrics presented in Figure 7 and Table 1. Only the manual comparison is shown visually in Figure 6 as a demonstration of the approach - this section has also been updated following comments from Reviewer 2 to reflect a more robust method for identifying the bed ridges and valleys from the radar data and matching them to mapped features.*

358 Modelling suggests that for ice sliding over the bed, bed topography is transferred to the ice surface. However, earlier in the paper you mention that the ice is most likely frozen to the bed, allowing it to preserve the pre-glacial topography. Does that mean that the ice is not sliding over the bed, and flowing slowly through internal deformation, and do you think that would have an influence on the way in which bed topography is transferred to the ice surface in this region?

*Yes, models consistently indicate that ice is predominantly frozen and not sliding (DeConto and Pollard, 2003; Creyts et al. 2014; Wolovick et al. 2021) – this does suggest that the mechanism for transferring the topographic signature to the ice surface must be different to the process observed in sliding ice. However the results indicate that ice moving predominantly through internal deformation is still affected by bed topography, especially when the bed relief is significant (c.f. Ross et al., 2014). Flow of ice under these conditions can be extremely complicated (Martinez, 2021) and modelling it is beyond the scope of this paper. Further understanding the processes involved would be an intriguing goal for future work. Nonetheless, the results indicate that certain findings of Ockenden (2021) are still pertinent, including the underrepresentation of landforms that are oriented parallel to modern ice flow in the ice surface morphology. We make it clear in the text that, while the*

*overriding principle is similar, the processes involved in the two different contexts may be different.*

**Technical comments**

79 I feel like this could be rephrased with fewer commas to read more smoothly. 'The geology of the Gamburtsev Subglacial Mountains is poorly understood, with their...'

*We rephrase the sentence accordingly.*

100 You could be more explicit about the directionality of the relationship between oxygen isotopes and glaciation here; 'as increased terrestrial ice volume leads to a higher d18O in seawater and hence in benthic organisms.'

*We include a short explanation of the link between ice volume and d18O as suggested.*

115 It is very difficult to see the surface curvature in Figure 2d which is overlain on the RAMP intensity. I would suggest plotting these two separately, or increasing the transparency of the RAMP Intensity layer.

*Figure 2 has been adapted to have an additional panel so that the RAMP intensity and REMA curvature datasets can each be shown separately.*

124 Bed elevation models? Or a bed elevation model?

*We rephrase this sentence to be specific about the datasets used: "... and assessed the results against the AGAP radar survey bed elevation data (Bell et al. 2011; Ferraccioli et al., 2011), and the BedMachine bed elevation model (Morlighem et al., 2020)."*

145 Figure 3, could you include a colour scale bar for mean curvature with this figure. If the scale is different for each plot, perhaps a normalised colour scale could be used.

*We will produce a version of this figure with a normalised scale and scale bar as suggested.*

169 REMAv2 has many fewer gaps than REMAv1, so I'm curious to know which version was used (there is no timestamp on the reference for the dataset). Do you think that these gaps made a significant impact to the results of your study?

*We indicate in the text that REMAv1 was used. We also include a statement of the data coverage for the study area to indicate that missing data had a minimal impact on the results (Data coverage over the study area is very good, with data in 99.92% of 200-by-200 m cells).*

209 It might be good to mention that the technique used by Bedmachine Antarctica to go from the AGAP RES data to their DEM is streamline diffusion (and not mass conservation as many readers might assume).

*We include this information as suggested e.g. "The BedMachine Antarctica version 2 bed DEM (Morlighem et al., 2020), which is primarily derived from the AGAP RES data in this area using the streamline diffusion technique (Morlighem et al., 2020), ..."*

**Reviewer #2 Comments**

Lea et al present a map of ridge and valley features based on satellite observations of the surface. They validate the inferences using the dense AGAP radar survey. This technique allows mapping beyond the radar survey, and, in theory, resolution at length scales shorter than the radar line spacing. The technique is clever and offers additional information to the radar data. The paper would be much improved from a more quantitative validation with objective definitions of ridges and valleys in the radar data as the qualitative assessment (i.e. Figure 6) is not particularly convincing. The technique, with additional quantitative validation, will be a useful addition to tools for inferring subglacial features in regions undersampled by radar data.

*We thank the reviewer for their positive comments on the technique developed in the paper, and acknowledge the need for a more robust procedure to quantitatively validate it with regards to objectively defined bed ridges and valleys. We address these concerns through an updated version of the procedure outlined in response to specific comments below.*

The technique is, in general, well described. It would be useful to expand Figure 4 to include both a valley as well as short length scale identifications of valleys and ridges, e.g. segment 10 10-16km. Figure 4 is a good example, but likely lacks many of the complications that arise between manual and automated interpretation.

*It is true that Figure 4 does not cover every eventuality when it comes to mapping by hand, as it is intended to be illustrative of the technique only. Following comments from Reviewer 1 also, we include a more detailed description of both the manual and automated methods as a supplement to the main paper to address this and similar concerns.*

The validation suffers from a lack of quantification of valley and ridges in the radar data. How a ridge and a valley are defined in the radar data is not clear. The authors provide statistics of the percentage of features that are correctly identified, but do not describe how the "true" features are defined. Figure 6 shows the inferred features, and the bed data, but not the "true" ridges and troughs defined by the radar data. Without a specific definition of ridges and troughs, quantitative metrics of how many features were correctly and incorrectly identified do not convince me. In Figure 6, segment 10, 5 features are identified between 10 and 16km, but the features are very small in scale and more significant features between 3 and 10 km are not identified.

Figure 6 also reveals that the same scales of basal features are not consistently identified between segments. For instance, small features are identified in segment 10, but are not identified in Figure 9. The bed elevation is similar between them, so it's unlikely that a greater ice thickness is obscuring the bed variations.

*We implement a modified validation procedure which identifies ridges and valleys as local maxima/minima in the radar data with minimum 100 m prominence. This excludes 41 "true" features originally identified qualitatively, reducing the sample size used in validation, but as the Reviewer notes, it makes the quantitative metrics more robust and meaningful. Figures 6 and 7 and Table 1 are updated accordingly. We also include additional discussion of the spatial variability in scale of features identified.*

It would be much more useful to have the "true" ridges and troughs marked, solid lines that indicate what the authors consider a successfully identified feature, and dashed lines for incorrect features. Further, the authors need to define a distance that the mapped feature must match the true feature – or possibly define a ratio of distance to amplitude of feature.

*An updated version of Figure 6 (now Fig. 7) following the modified validation procedure adopts these suggestions - "true" bed features are marked using red and blue triangles, matched features using solid lines, unmatched features as dashed lines. The distance cut-off for matching features was set at 2 km (as very few matched features were more than 2 km distant from the "true" feature in the original validation procedure).*

It is surprising to me how often the same features are identified successively (i.e. 2 or more ridges in a row). For instance, Figure 6, segment 9 has 3 ridges between 5 and 10 km without any valleys in between. And then the only valley is identified at a ridge at just over 15km. It seems like a sequence of multiple of the same feature in a row would be a good check that could be implemented into the automated routine to help improve its performance.

*This occurs mainly as a function of sampling the two-dimensional planform network along the discrete flightlines traced by the radar data. Each vertical line in Figure 6 (now Fig. 7) represents a point where the flightline intersects a mapped ridge or valley. Because ridges and valleys are generally only digitised as far as they are visible in the datasets used for mapping (RAMP, REMA curvature), there is no guarantee of a strict ridge-valley-ridge progression along a given flightline. This is the case, for instance, where flightlines cross the upper portions of valleys (where the valley may not have been digitised all the way to the valley head) or broader valleys close to confluences (where there may not be a discernible ridge between channels). A more detailed explanation of the updated validation procedure is included to make this sampling relationship more explicit.*

*The automated routine (as detailed in the included supplement) evaluates on a pixel-by-pixel basis whether a pixel is identified as a ridge or valley by its relationship to its immediate neighbours, and does not intrinsically consider proximity to other features in any given direction. We include a sentence in the supplement to say that such a check would be beneficial in a more advanced version of the procedure, but note that it is not straightforward to implement in two dimensions and so is not included in this paper. We also include a sentence in the main text which summarises this procedure.*

*We also note that the instances identified by the Reviewer in Figure 6 (now Fig. 7) are all examples from manual mapping only, and that this has been made clearer.*

I would also like to see a more quantitative analysis of the acceptable distance between a "true" and a mapped feature. There are many instances in Figure 6 where there are potentially multiple features in the data that the mapped feature could be assigned to.

*The modified validation algorithm assigns a bed feature to the nearest mapped feature that is not closer to a different bed feature (to which it is therefore assigned). No more than one mapped feature is assigned to each bed feature, and no mapped feature is matched if it is*

*more than 2 km distant from the nearest bed feature. Note that this procedure is limited by the one-dimensional nature of the radar data being used for comparison – it is entirely possible that unmatched features in Figure 6 represent ridges or valleys with offsets in a lateral direction (i.e. into or out of the page rather than along the flightline). The variability in offset distance and direction appear to be related to factors including local variability in ice flow, as mentioned in the discussion in this section, and indicated in the updated version of Figure 6 (now Fig. 7).*

While most of my comments have to do with the validation, there are a couple of other areas where the paper could be improved. A morphological analysis of a few alpine regions that could be directly compared with GSM would be very informative. How do the Rockies compare quantitatively? Is GSM closer to the northern Rockies (i.e. Glacier NP) or the southern Rockies (ie. The San Juans in CO). Or I'm sure you come up with better analogs to determine the extent of glaciation that may have occurred.

*Assuming this comment refers to the morphometric analyses of the valley network, it is possible to replicate some of the analyses performed by using valley networks derived from DEMs, such as the global dataset constructed by Lin et al. (2021). The primary metric used in the paper that could be compared quantitatively is ridge-to-ridge valley width (valley spacing) – our original intention was mainly to demonstrate using this analysis that valleys in the GSM have widths smaller than those identifiable using other methods, and that these widths are on par with those found in well-known alpine environments. To this end the paper does reference observations by Pelletier et al. (2010) of typical first-order valley widths in three North American ranges (namely, the Beartooth-Absaroka Mountains, Montana; the Uinta Mountains, Utah; the central Rocky Mountains, Colorado). They note that these lie generally in the range 1-3 km, which accords well with the range displayed in Figure 9 of this paper for the GSM. Given that the method applied here is only a proxy for true valley widths, which can be measured directly in other ranges from a surface DEM, it seems unlikely that any more specific comparison could be made based on applying it in these locations. Therefore, in order to avoid reprising analyses that have previously been conducted by other researchers, we add detail to the comparison with the observations of Pelletier et al. (2010) and suggest the ranges studied in this previous work as potential analogues. We also include as part of the online repository the GIS workflow used to create the raster of mean valley spacing used in Figure 9 (now Fig. 10), so that it can more easily be applied to other vector representations of valley networks (e.g. Lin et al., 2021)*

*We are sceptical that it is feasible to determine the extent of a past glaciation in the GSM using only the planform networks derived in this study, and note that this was never an intended aim of the paper.*

**Additional Comment (received via email)**

I'm writing to let you know I really enjoyed reading your Discussion paper in TC on alpine topography of the GSM. It's very well presented and well written, so congratulations for bringing together a focused new study on the topic.

As you take in outside review input, I have just a couple of suggestions for minor revisions or discussion points:

Fig. 1 — AGAP not shown on figure, which corresponds to unlabeled box with dashed outline?

*The dashed box corresponds to the central mesh of the AGAP Survey. This is indicated in the Figure legend and is now made clearer with a label on the Figure itself.*

Fig. 2b, label ice catchments.

*Labels have been added.*

I'm attaching a couple of papers that are relevant to your discussion of GSM origin scenarios and might be of interest. Together these address origins of igneous crust in central East Antarctica (Goodge et al., 2017) and its thermal evolution during the Paleozoic to Cenozoic (Fitzgerald and Goodge, 2022). These studies of glacially-eroded clasts may, in contrast to the cited studies from debris in Prydz Bay, give geological clues to the composition and age of the enigmatic GSM. In particular, material sourced from the southern flank of the GSM and transported into the Byrd Glacier drainage have Proterozoic igneous ages and cooling ages ≤500 Ma that together point to a rather ancient origin. To be sure, we don't know how far these cobbles were transported, so their source might not be as far as the GSM, but they seem to provide the most tangible geological evidence for crust in the region, particularly as their igneous ages are largely unknown elsewhere in exposed parts of Antarctica, and they preserve a long cooling and exhumation history.

*Key findings from these papers are included in the discussion of Gamburtsev age and origin in section 1.1 and/or later in the paper.*

**Editor Comments**

L8 It is not clear if "this time" refers to prior to and during the glaciation or since the last glaciation. Please modify the sentence to clarify.

*Sentence updated to refer specifically to time since ice sheet inception.*

L28 In general the in-text citation format used is by chronological order. Therefore, please modify this reference to be consistent with other in-text citations (chronological order)

*Reference order updated accordingly.*

L397 see comment for L28

*Reference order updated accordingly.*

**Other**

Line 318 – wrong figure reference, should say Fig. 10 (was Fig. 9) at end of line, not Fig. 5.7 – this has been updated.

**References**

Creyts, T.T. et al. (2014) 'Freezing of ridges and water networks preserves the Gamburtsev Subglacial Mountains for millions of years', Geophysical Research Letters, 41(22), pp. 8114–8122. Available at: https://doi.org/10.1002/2014GL061491.

DeConto, R.M. and Pollard, D. (2003) 'Rapid Cenozoic glaciation of Antarctica induced by declining atmospheric CO2', Nature, 421(6920), pp. 245–249. Available at: https://doi.org/10.1038/nature01290.

Fitzgerald, P.G. and Goodge, J.W. (2022) 'Exhumation and tectonic history of inaccessible subglacial interior East Antarctica from thermochronology on glacial erratics', Nature Communications, 13(1), p. 6217. Available at: https://doi.org/10.1038/s41467-022-33791-y.

Goodge, J.W. et al. (2017) 'Proterozoic crustal evolution of central East Antarctica: Age and isotopic evidence from glacial igneous clasts, and links with Australia and Laurentia', Precambrian Research, 299, pp. 151–176. Available at: https://doi.org/10.1016/j.precamres.2017.07.026.

Lin, P. et al. (2021) 'A new vector-based global river network dataset accounting for variable drainage density', Scientific Data, 8(1), p. 28. Available at: https://doi.org/10.1038/s41597-021-00819-9.

Martinez, E. (2021) 'Moffatt Eddies in Subglacial Mountain Valleys', ENGS 88 Honors Thesis (AB Students) [Preprint]. Available at: https://digitalcommons.dartmouth.edu/engs88/28.

Ockenden, H. et al. (2021) 'Inverting ice surface elevation and velocity for bed topography and slipperiness beneath Thwaites Glacier', The Cryosphere Discussions, pp. 1–34. Available at: https://doi.org/10.5194/tc-2021-287.

Pelletier, J.D., Comeau, D. and Kargel, J. (2010) 'Controls of glacial valley spacing on earth and mars', Geomorphology, 116(1), pp. 189–201. Available at: https://doi.org/10.1016/j.geomorph.2009.10.018.

Ross, N. et al. (2014) 'The Ellsworth Subglacial Highlands: Inception and retreat of the West Antarctic Ice Sheet', GSA Bulletin, 126(1–2), pp. 3–15. Available at: https://doi.org/10.1130/B30794.1.

Wolovick, M.J., Moore, J.C. and Zhao, L. (2021) 'Joint Inversion for Surface Accumulation Rate and Geothermal Heat Flow From Ice-Penetrating Radar Observations at Dome A, East Antarctica. Part II: Ice Sheet State and Geophysical Analysis', Journal of Geophysical Research: Earth Surface, 126(5), p. e2020JF005936. Available at: https://doi.org/10.1029/2020JF005936.